


# Investigating the Source, Transport, and Isotope Composition of Water Vapor in the Planetary Boundary Layer

T.J. Griffis[1], J.D. Wood[1], J.M. Baker[1,2], X. Lee[3], K. Xiao[1], Z. Chen[1], L.R. Welp[4], N. Schultz[3], G. Gorski[1], M. Chen[1], and J. Nieber[5]

[1]Department of Soil, Water, and Climate, University of Minnesota, Saint Paul, MN, USA
[2]United States Department of Agriculture - Agricultural Research Service, Saint Paul, MN, USA
[3]School of Forestry and Environmental Studies, Yale University, New Haven, CT, USA
[4]Earth, Atmospheric, and Planetary Sciences, Purdue University, West Lafayette, IN, USA
[5]Department of Bioproducts and Biosystems Engineering, University of Minnesota, Saint Paul, MN, USA

*Correspondence to:* Tim Griffis (timgriffis@umn.edu)

**Abstract.**

Increasing atmospheric humidity and convective precipitation over land provide evidence of intensification of the hydrologic cycle – an expected response to surface warming. The extent to which terrestrial ecosystems modulate these hydrologic factors is important to understanding feedbacks in the climate system. We measured the oxygen and hydrogen isotope composition of water vapor from
a very tall tower (185 m) in the Upper Midwest, United States to help diagnose the sources, transport, and fractionation of water vapor in the planetary boundary layer (PBL) over a 3-year period (2010 to 2012). These measurements represent the first set of annual water vapor isotope observations for the region. Models and cross wavelet analyses were used to assess the importance of Rayleigh, evapotranspiration (ET), and PBL entrainment processes on the isotope composition of water va-
por. The vapor isotope composition at this tall tower site showed a very large seasonal amplitude (mean monthly $\delta^{18}O_v$ ranged from -40.1 to -15.5‰ and $\delta^2H_v$ ranged from -278.7 to -109.1‰) and followed the familiar Rayleigh distillation relation with water vapor mixing ratio at the annual time-scale. However, this relation was strongly modulated by ET and PBL entrainment processes
at time-scales ranging from hours to several days. The wavelet coherence spectra indicate that the oxygen isotope ratio and the deuterium excess ($d_x$) of water vapor are sensitive to synoptic and PBL processes. According to the phase of the coherence analyses, we show that ET often leads changes in $d_x$, confirming that it is a potential tracer of regional ET. Isotope mixing models indicate that on average about 31% of the growing season PBL water vapor is derived from regional ET. However,
isoforcing calculations and mixing model analyses for high PBL water vapor mixing ratios events ($> 25$ mmol mol$^{-1}$) indicate that regional ET can account for 40% to 60% of the PBL water vapor. These estimates are in relatively good agreement with that derived from numerical weather model



simulations. This relatively large fraction of ET-derived water vapor implies that ET has an important impact on the precipitation recycling ratio within the region. Based on multiple constraints, we

estimate that the summer season recycling fraction is about 30%, indicating a potentially important link with convective precipitation.

## 1  Introduction

There is unequivocal evidence that the global water cycle has been intensified by anthropogenic warming (Chung et al., 2014; Trenberth et al., 2007a; Santer et al., 2007). Global analyses demon-

strate that water vapor is increasing over the oceans (Santer et al., 2007), at continental locations (Dai, 2006), and in the upper troposphere (Chung et al., 2014). Quantifying and elucidating the processes underlying the variability in atmospheric water vapor remains one of the grand challenges in water cycle science (Trenberth and Asrar, 2014).

Higher water vapor concentrations are expected to have important impacts on climate (Trenberth

et al., 2007a). Water vapor is the dominant greenhouse gas, accounting for about 50% of the longwave radiative forcing (Schmidt et al., 2010), and also plays a key role in atmospheric aerosol formation (Nguyen et al., 2015) and therefore short-wave radiative forcing. Furthermore, water vapor is an active scalar influencing static stability and convection. There is growing evidence that the frequency and magnitude of convective precipitation events are increasing as a result of surface warming and

higher humidity (Trenberth et al., 2007a; Trenberth, 2011; Min et al., 2011).

Interpreting the variations in water vapor over continental locations is challenging because there are many different sources, transport processes, and phase changes that influence water vapor history on a variety of temporal and spatial scales. In recent years there have been important technical advances that have enhanced our ability to quantify the oxygen ($\delta^{18}$O) and deuterium ($\delta^{2}$H) isotope

composition of water vapor and evapotranspiration (ET) using optical isotope techniques (Lee et al., 2005; Wen et al., 2008; Welp et al., 2008; Wang et al., 2010; Johnson et al., 2011; Noone et al., 2013; Griffis, 2013). These technical advances are now providing high density datasets that can be used to diagnose how hydro-meteorological factors (i.e. air mass back trajectories, precipitation, lake evaporation, and snow sublimation) (Lee et al., 2006; Noone et al., 2013; Farlin et al., 2013; Soderberg

et al., 2013; Delattre et al., 2015) and biophysical factors (i.e. transpiration, soil evaporation) (Welp et al., 2008; Hu et al., 2014; Simonin et al., 2014) influence land-atmosphere water vapor exchange and the sources of water contributing to atmospheric water vapor.

The isotope composition of water vapor in the planetary boundary layer (PBL) can vary strongly on seasonal and diurnal time scales depending on geographical location (Welp et al., 2012). Diurnal

variations have been linked to PBL entrainment processes (Lai and Ehleringer, 2011; Lee et al., 2012; Welp et al., 2012; Noone et al., 2013) and ET (Lee et al., 2007; Griffis et al., 2010b; Lai and Ehleringer, 2011; Welp et al., 2012; Huang and Wen, 2014). There is growing consensus that





water vapor deuterium excess ($d_x = \delta^2$H- $8\delta^{18}$O) is not a conserved quantity of marine evaporation conditions as once thought, but that it is highly sensitive to changes in ET and PBL processes (Welp et al., 2012; Zhou et al., 2014; Huang and Wen, 2014). The high sensitivity of isotopes in water vapor, $\delta^2$H$_v$, $\delta^{18}$O$_v$, and $d_x$ to ET may, therefore, offer new insights regarding the controls and water sources influencing continental atmospheric water vapor and precipitation.

Here, we examine the temporal scales and extent to which Rayleigh, ET, and PBL growth processes influence the isotope compositions ($\delta^2$H$_v$, $\delta^{18}$O$_v$, and $d_x$) of mid-continental atmospheric water vapor as observed in the Upper Midwest United States. We then use these tracers to help constrain the precipitation recycling fraction at the tall tower site. Figure 1 provides an overview of our investigation and illustrates the spatial domain and methodological approach. We bring together an unique multi-year (2010-2012) record of tall tower water vapor mixing ratio (major and minor isotopes), precipitation isotope ratios (2006-2011), surface vapor flux observations, cross-wavelet analyses, and numerical modeling to evaluate the following hypotheses:

1. The isotope composition of the PBL within this region is largely determined by air mass Rayleigh distillation, but is strongly modulated by ET at time-scales ranging from hours to days.

2. The deuterium isotope signal in PBL water vapor is most strongly influenced by regional ET.

3. The growing season water vapor concentration in the PBL is dominated by regional ET from crop lands.

4. Growing season precipitation events are comprised of a significant contribution of local ET and therefore exhibit a relatively high degree of moisture recycling.

## 2 Methodology

### 2.1 Study Site

The measurements reported in this study were made at the University of Minnesota tall tower trace gas observatory (KCMP, Minnesota Public Radio tower, 290 m ASL, $44°41'19''$ N, $93°4'22''$ W). The tall tower (244 m) is located about 25 km south of Saint Paul, Minnesota (Figure 1). It was instrumented in spring 2007 with air sample inlets at 32, 56, 100, and 185 m. Three-dimensional sonic anemometer-thermometers (CSAT3, Campbell Scientific Inc., Logan, Utah, USA) are mounted at 100 m and 185 m, with signals transmitted to data loggers and computers via fiber optic cables and modems (Griffis et al., 2010a). Scalars including carbon dioxide, water vapor, nitrous oxide, methane, isoprene,and other trace gases have been measured at the site since 2007 (Griffis et al., 2010a, 2013; Hu et al., 2015a, b). Land use in the vicinity of the tall tower (extending from 10 to 600 km radius) consists of about 40% agriculture (mainly corn and soybean) that is typical of the



US corn belt (Griffis et al., 2013; Zhang et al., 2014). The concentration footprint of the tall tower (185 m sample inlet) when coupled to inverse model analyses has shown to be representative of the Upper Midwest United States for a number of active and passive scalars (Zhang et al., 2014; Hu et al., 2015b).

## 2.2 Isotope Measurements

The oxygen and hydrogen isotopes in water vapor were measured *in situ* using a tunable diode laser (TDL) (model TGA200, Campbell Scientific Inc., Logan, Utah, USA) (Lee et al., 2005; Griffis et al., 2010b). These measurements were initiated April 2010. Air was pulled down sample tubing at the TGO to the analyzer that was maintained inside the climate controlled radio broadcast building.

The sample inlets used in this investigation were located at approximately 185 m and 3 m above the ground surface. The tubing was heated from the base of the tower to the laser sample inlet, a distance of about 30 m, to prevent condensation. The sampling scheme consisted of a 10-min (600 s) cycle: (1) zero calibration with ultra dry air (110 s), (2) calibration with three span values (15 s/each) for the 3 m inlet, (3) sampling of the 3 m inlet (145 s), (4) zero calibration with ultra dry air (110 s),

(5) calibration with three span values (15 s/each) for the 185 m inlet, and (6) sampling of the 185 m inlet (145 s). An omit time of 5 s was used on the calibration spans and air samples, and a 90 s omit time was used for the dry air calibration. All raw data were recorded at 10 Hz using a data logger and then block-averaged into one hour intervals. The hourly water vapor signals were filtered using an outlier detection algorithm based on the double-differenced time series that identifies outliers ac-

cording to the median absolute deviation about the median values (Sachs, 1996; Papale et al., 2006). Further details regarding the post-processing calibration techniques and uncertainties are described in (Griffis et al., 2010b).

Precipitation samples have been collected from RROC, and at the University of Minnesota-Saint Paul campus from January, 2006 to present using a typical all-weather rain gauge with mineral oil

added to eliminate evaporative fractionation effects. Samples were typically collected within 0-3 days of precipitation events and transferred to screw-top glass vials, sealed with Parafilm and refrigerated until analysis. The timing and amount of rainfall was recorded using a tipping bucket rain gauge, and snowfall was measured using a snow board. Leaf, stem, and soil samples were collected from within a 5 km radius of the tall tower during numerous campaigns and as part of the Moisture

Isotopes in the Biosphere and Atmosphere (MIBA) program (http://www-naweb.iaea.org/napc/ih/IHS resources miba.html). Vegetation sampling sites chosen for this analysis were representative of the local land cover characteristics including corn (*Zea mays* L.), soybean (*Glycine max*), and big bluestem (*Andropogon gerardii Vitman*). The MIBA sampling protocol was followed. Sunlit leaves, non-green stems, and soil approximately 10 cm below the surface were collected near midday (1200

local standard time (LST)). Cryogenic vacuum distillation (Welp et al., 2008; Schultz et al., 2011)





was used to extract water from the plant and soil samples. Surface (i.e. lake and river) water and ground water samples were also collected from within a 25 km radius of the tall tower.

All liquid water samples were analyzed for their isotope composition using an off-axis cavity ring-down infrared laser spectroscopy system (Liquid Water Isotope Analyzer, DLT-100, Los Gatos Research, Inc., Mountain View, California) coupled to an autosampler (HT-300A, HTA s.r.l., Brescia, Italy) for simultaneous measurements of $^2$H/$^1$H and $^{18}$O/$^{16}$O. This instrument has a precision of $\pm$ 1.0‰ for $^2$H/$^1$H and $\pm$ 0.25‰ for $^{18}$O/$^{16}$O. Pre-calibrated laboratory standards used to calibrate the unknown samples to the VSMOW-PDB scale were selected based on the expected isotope composition of the unknown samples, and were injected after every two unknown samples to correct for instrumental drift. Linear calibration equations were calculated using each set of standards throughout the autorun and used to correct unknown samples. Contamination of plant water samples by ethanol/methanol were corrected following the procedures described by Schultz et al. (2011).

### 2.3 Wavelet Analyses

Signals were analyzed using techniques based on the continuous wavelet transform (CWT). Wavelet based techniques are particularly suited to analyzing non-stationary geophysical time series because signals are simultaneously decomposed into time-frequency space. See Daubechies (1990) and Torrence and Compo (1998) for an overview of the theoretical background and practical application. Here, we use cross wavelet analyses to help elucidate how different atmospheric processes influence the isotope composition of PBL water vapor and to better understand the patterns and timescales of those relations.

Briefly, all CWT's were calculated on the fluctuating component of the signal using the complex Morlet wavelet basis with the nondimensional frequency ($\omega_0$) set to 6 (Torrence and Compo, 1998) to obtain a good balance between time and frequency localization (Grinsted et al., 2004). Another desirable feature of the Morlet wavelet basis with $\omega_0 = 6$ is that the scales map closely to an analogous Fourier period ($\lambda$) according to: $\lambda = 1.03s$ (Torrence and Compo, 1998), where $s$ is the scale, and the dimension of both $\lambda$ and $s$ is time. Scales were set to have a minimum of 2 h (i.e. twice the hourly averaging interval), and to have 12 sub-octaves per octave. Calculating the CWT of the signal yields a set of wavelet coefficients, $W_n(s)$, spanning all times ($n$) and scales. Here, we concern ourselves with disentangling the effects of different processes on PBL water vapor, and thus employ the multivariate technique known as wavelet coherence analysis to probe correlation and phase relationships between variables.

The cross wavelet spectrum, $S_n^{XY}(s)$, of two time series, $x_n$ and $y_n$, is obtained from the wavelet coefficients calculated for the respective variables according to:

$$S_n^{XY}(s) = W_n^X(s)W_n^Y(s)^* \tag{1}$$





where $*$ represents complex conjugation (Grinsted et al., 2004). The cross wavelet spectrum iden-
tifies regions of high common power, but does not provide information regarding the coherency
between the signals.

To examine the coherency of the cross wavelet transform in time frequency space, we made use
of the wavelet coherence spectrum, $R_n^2(s)$, that is defined according to:

$$R_n^2(s) = \frac{|\Lambda(s^{-1}S_n^{XY}(s))|^2}{\Lambda(s^{-1}|S_n^X(s)|^2)\Lambda(s^{-1}|S_n^Y(s)|^2)} \qquad (2)$$

where $\Lambda$ represents a smoothing operator and its definition can be found in Grinsted et al. (2004) (see
their equations 9 and 10). A useful interpretation of the coherence spectrum is that values of $R_n^2(s)$
represent local correlation coefficients in time-frequency space (Grinsted et al., 2004). Statistical
significance testing was performed using the Monte Carlo approach described in Grinsted et al.

(2004). All wavelet analyses were implemented using the package of MATLAB functions developed
by Grinsted et al. (2004), which is available at http://www.glaciology.net/wavelet-coherence.

### 2.4    Numerical Modeling

We used the National Center for Atmospheric Research (NCAR), Weather Research and Forecast-
ing (WRF) model version 3.5 to simulate the regional surface latent heat flux, PBL height, and

to examine other controls on the regional water vapor (Chen et al., 1996). The simulations made
use of 4 nested domains (with a recommended 3:1 ratio for inner domains) with the inner-most
domain containing the location of the tall tower. The inner domain 4 occupied the smallest area
(80 x 80 km) and employed a 1 km grid resolution (see Figure A2 in the auxiliary file). In these
simulations a 2-way feedback among the nested domains was turned on. The NOAH land sur-

face scheme option was selected for all WRF simulations. The WRF-NOAH simulations used land
surface information from the United States Geological Survey (USGS) land use product, which
includes 24 land use categories. The WRF settings (namelist file) used to run these simulations
are provided in the auxiliary information. Boundary and initial conditions were provided by the
NCEP FNL Operational Global Analysis data product with a $1^\circ$ x $1^\circ$ resolution at 6 hour intervals

(http://rda.ucar.edu/datasets/ds083.2/). Further, the Stochastic Time-Inverted Lagrangian (STILT)
model (Lin et al., 2003; Gerbig et al., 2003) was used to examine the water vapor concentration
source footprint associated with an extreme dew point event at the tall tower. The meteorological
fields required to drive STILT were obtained from the WRF simulations. Since water vapor is an ac-
tive scalar, the STILT source footprints computed here likely represent the maximum spatial extent

of influence with respect to the tall tower observations. All of these model simulations were run on
an HP ProLiant BL280c G6 Linux Cluster at the University of Minnesota Supercomputing Institute
(https://www.msi.umn.edu/).





### 2.5 Basic Isotope Theory

The isotope composition of precipitation and water vapor is reported as,

$$\delta = \frac{R_s - R_{std}}{R_{std}} \qquad (3)$$

where $\delta$ is the isotope ratio. All values are reported in parts per thousand (‰) by multiplying $\delta$ by $10^3$. $R_s$ is the sample molar ratio of the heavy (minor) to light (major) isotope (i.e. $^{18}O/^{16}O$ or $^2H/^1H$) and $R_{std}$ is the standard molar ratio defined according to the V-SMOW scale.

We make use of precipitation events to examine the isotope composition of water vapor in relation to the falling precipitation. In theory, if atmospheric humidity is at saturation below the cloud base, then thermodynamic equilibrium is expected for isotope exchange between the liquid water and atmospheric vapor (Stewart, 1975),

$$R_v = R_L/\alpha \qquad (4)$$

where $R_v$ is the absolute isotope ratio of water vapor ($^{18}O/^{16}O$ or $^2H/^1H$), $\alpha$ is the equilibrium fractionation factor (isotope specific), and $R_L$ is the isotope ratio of the liquid water (rain precipitation) (Lee et al., 2005). Under these conditions, the equilibrium relation can provide a useful diagnostic regarding the validity of the tall tower water vapor isotope ratios or the influence of evaporation of raindrops and humidification of the PBL.

The global meteoric water line (GMWL),

$$\delta^2H = 8\delta^{18}O + 10 \qquad (5)$$

represents the linear relation between $\delta^2H$ and $\delta^{18}O$ for global precipitation and is a useful benchmark for examining the origin, modification, and history of other water sources (Craig, 1961; Gat, 1996). The GMWL parameters are derived from empirical observations and are related to Rayleigh distillation processes (Gat and Airey, 2006). The slope of $\approx 8$ results from the equilibrium condensation conditions and the ratio of the equilibrium fractionation factors (Jouzel, 2003). The intercept of $\approx 10$ is determined by the average equilibrium and kinetic fractionation factors for ocean-atmosphere exchange with a global evaporation-weighted mean relative humidity of $\approx 85\%$ (Clark and Fritz, 1997). Sources of water undergoing evaporation result in isotope kinetic effects that cause $\delta^2H$ -$\delta^{18}O$ slopes less than 8 (Dansgaard, 1964; Gat et al., 1994; Gat and Airey, 2006).

Three simple models were used to aid the interpretation of the tall tower $\delta^{18}O_v$ data. First, a Rayleigh model (RM1) assuming a closed system with no rain out was assessed (Lee et al., 2006),

$$\delta_{RM1} = 1000(\alpha - 1)(log(\chi_w) - log(\chi_o)) + \delta_o; \qquad (6)$$





where $\alpha$ is the equilibrium fractionation factor evaluated at a condensation temperature of -3°C (this represents the mean adiabatically-adjusted temperature at the lifted condensation level). Here, the initial air mass is assumed to have an oceanic source region with a water vapor mixing ratio ($\chi_o$) of 35 mmol mol$^{-1}$ and an oxygen isotope ratio ($\delta_o$) of -10‰ (Worden et al., 2007). While these initial values are somewhat arbitrary, it is the variation in the response function relative to the observations that is of primary interest. Second, a Rayleigh model (RM2) with a rain-out fraction ($f$) of 30% was evaluated,

$$\delta_{RM2} = 1000(\alpha(1-f/\alpha)/(1-f) - 1)(log(\chi_w) - log(\chi_o)) + \delta_o; \qquad (7)$$

where precipitation/condensation is evaporated causing the isotope composition of the water vapor to become relatively more depleted (Worden et al., 2007; Lee et al., 2006). Finally, a simple two-source evaporation mixing model (EM1, a Keeling plot, (Keeling, 1958)) was examined,

$$\delta_{EM1} = \frac{\chi_b}{\chi_w}(\delta_b - \delta_{\text{ET}}) + \delta_{\text{ET}} \qquad (8)$$

that considers surface evaporation into an air mass. $\chi_w$ and $\chi_b$ represent the air mass and background water vapor mixing ratios, respectively. Here, the oxygen isotope ratio of evaporation ($\delta_{\text{ET}}$) is taken as -7.7‰, which is based on oxygen isotope eddy covariance measurements taken over a corn canopy (Griffis et al., 2010b).

The isoforcing ($I_F$) approach (Lee et al., 2009; Griffis et al., 2010a) was used to help interpret short-term (hourly) variations in the water vapor isotope observations,

$$I_F = \frac{\text{ET}}{C_a}(\delta_{\text{ET}} - \delta_v) \qquad (9)$$

where $C_a$ is the molar density of water vapor, $\delta_{\text{ET}}$ is the oxygen isotope composition of evapotranspiration as determined from the tall tower flux-gradient measurements (Schultz, 2011), and $\delta_v$ is the oxygen isotope composition of the water vapor in the PBL. The $I_F$ calculations are used to isolate the influence of ET on $\delta_v$. Although the same approach can be applied using the deuterium isotopes, the atmospheric gradients are considerably smaller, resulting in low signal to noise ratios. As a result, we restricted our deuterium isoforcing calculations to the mid growing season (may through August).

A simple two-member isotope mixing model was used to estimate the relative contribution of ET to the total water vapor concentration of the PBL,

$$f_v = \frac{\delta_v - \delta_b}{\delta_{\text{ET}} - \delta_b} \qquad (10)$$

where $f_v$ is the fraction of vapor in the PBL derived from local evaporation, $\delta_v$ is the oxygen isotope composition of the water vapor measured at 185 m, and $\delta_b$ represents the oxygen isotope ratio





of the "background" vapor. Direct observations of the oxygen isotope composition of background

vapor for the region do not exist. However, we make use of an unique set of aircraft observations
collected by He and Smith (1999) over New England, USA in 1996. They obtained profiles of water
vapor mixing ratio and $\delta^{18}O_v$ at altitudes ranging from 195 m to 2851 m during three campaigns
(June 15, 1996, July 17, 1996, and October 12, 1996). We have plotted their data in Figure 2 and
demonstrate that $\delta^{18}O_v$ follows a power law (Rayleigh) function with respect to water vapor mixing

ratio ($y = ax^b$, where $x$ is water vapor mixing ratio, $r^2 = 0.98$, $p < 0.001$, $n = 24$) through the PBL.
Here, we define the background signal assuming a power law relation for the tall tower site. In this
approach, the theoretical background value was obtained by evaluating the power law relation with
water vapor mixing ratio observed at 700 hPa (i.e. above the PBL at a standard atmosphere height of
approximately 3000 m).

Constraints on the oxygen isotope composition of ET ($\delta_{ET}$) were provided from multiple studies
conducted near the tall tower. The oxygen isotope composition of ET was determined over a corn
canopy using the eddy covariance approach (Griffis et al., 2010b, 2011). These studies showed that
$\delta_{ET}$ ranged from -20 to -5‰ with a mean flux-weighted value of -7.7‰ for a 74-day period in 2009.
The $\delta_{ET}$ of soybean crops has also been estimated within the study domain using the flux-gradient

approach (Welp et al., 2008) with values ranging from about -30 to +20‰ with a mean flux-weighted
value of -4.8‰ over the period June to September in 2006. Regional $\delta_{ET}$ has also been obtained
from our tall tower flux-gradient observations. These values were similar to those reported for the
above field-scale investigations with a mean flux-weighted value of -6.8‰ for the 2010 to 2012
growing season (Table 1). Further, based on plant stem water extractions, and assuming steady-state

conditions for the mid to late afternoon period, the oxygen isotope composition of transpiration can
be approximated as stem water (Welp et al., 2008). Our data from plant sampling in the vicinity of
the tall tower indicate a mean stem water oxygen isotope composition of -7.0‰ in 2010 (**?**).

Following the methodology of Gat et al. (1994) we estimated the recycling ratio of growing season
precipitation ($f_p$) using the two-member mixing model approach,

$$f_p = \frac{\delta_x - \delta_{adv_x}}{\delta_{ET_x} - \delta_{adv_x}} \qquad (11)$$

where, $\delta_x$ is the deuterium excess of precipitation, $\delta_{adv_x}$ is the deuterium excess of the advected
moisture (approximated here by the large concentration footprint of the tall tower water vapor mea-
surements at 185 m), and $\delta_{ET_x}$ is the deuterium excess estimated from the flux ratio measurements
at the tall tower.





### 285    3    Results and Discussion

#### 3.1    Isotope composition of water vapor in the PBL

Here we describe the climatology of the isotope composition of precipitation, water vapor, and ET as observed at the tall tower (Table 1). The mean oxygen and hydrogen isotope composition of precipitation (weighted by amount) was -11.5 and -80.1‰, with a range of monthly means of 14.2 and

123.3‰, respectively. The isotope signature of precipitation showed peak enrichment of the heavier isotopes in August. The mean deuterium excess of precipitation was 11.6‰ with a range of 12.5‰. Peak values were observed during November. The oxygen and hydrogen isotope composition of water vapor ($\delta^{18}O_v$ and $\delta^2H_v$) measured at the 185 m level had a mean annual value of -26.1 and -175.4‰, with a range of monthly means of 24.6 and 169.6‰, respectively. The isotope signature

of water vapor showed relatively strong enrichment of the heavier isotopes in July when the water vapor mixing ratio reached its maximum value. The mean annual deuterium excess ($d_x$) of water vapor was 33.5‰, with a range of 29.5‰. Deuterium excess of water vapor reached a minimum value in July.

The mean annual flux-weighted oxygen isotope ratio of ET ($\delta_{ET}$ = -13.2‰) was in excellent

agreement with the mean annual oxygen isotope ratio of the precipitation. There was strong seasonal variability in $\delta_{ET}$, with a mean growing season value of -6.0‰ over the 2010 to 2012 period, which was within the uncertainty of the oxygen isotope ratio of precipitation for the same period. The mean deuterium isotope composition of ET was -77.6‰ and was relatively depleted compared to precipitation. The effect of ET on the $\delta^{18}O_v$ and $\delta^2H_v$ of the PBL was estimated using the isoforc-

ing approach. The oxygen ET isoforcing was relatively strong from April to September with a mean value of 0.0068 m s$^{-1}$‰ (Table 1). The mean deuterium isoforcing was 0.004 m s$^{-1}$‰ from May through August. These calculations show that ET acts to enrich PBL water vapor in the heavier isotopes. We hypothesize that this contributes to the highly enriched values of convective precipitation observed during the growing season (discussed further below).

The observations reported here are in broad agreement with previous work conducted near Beijing, China. Wen et al. (2010) reported monthly water vapor mixing ratios that are in excellent agreement with the pattern and magnitude reported in Table 1. Further, their mean monthly values of $\delta^{18}O_v$ and $\delta^2H_v$ showed a peak in June and were -14.0 and -106.0‰, respectively. The deuterium excess in water vapor reached a minimum value of 5.7‰ in June. The $\delta^{18}O_v$ pattern and values reported

here are also similar to that observed near New Haven and Great Mountain Forest, Connecticut, USA (Lee et al., 2006). However, the continental location of Saint Paul, Minnesota exhibits a larger seasonal amplitude of $\delta^{18}O_v$ associated with the Rayleigh distillation effect, and perhaps, higher rates of ET and isoforcing from crops during the mid growing period.

The observed isotope ratios in water vapor, $\delta^{18}O_v$ and $\delta^2H_v$, measured at 3 m and 185 m were

compared with those derived from the isotope equilibrium theory ($\delta^{18}O_{v,e}$ and $\delta^2H_{v,e}$) for individual





precipitation events to gain insights regarding the validity of the tall tower observations and the isotope fractionation of water vapor in the PBL. Figure 3 shows results for 35 rain events from the 2010 to 2011 growing seasons. Overall, there was good agreement between the measured isotope ratios in water vapor compared to those predicted from the equilibrium theory. The mean measured

$\delta^{18}O_v$ was depleted by $1.38 \pm 0.38‰$ (uncertainty reported as the standard error) relative to the rain event $\delta^{18}O_{v,e}$ values. The linear regression shown in Figure 3a ($y = 0.54x - 7.3$, $r^2 = 0.42$, $p < 0.001$) supports that the derived equilibrium vapor values were biased low. A similar relation was observed for $\delta^2H_v$ ($y = 0.73x - 33.3$, $r^2 = 0.50$, $p < 0.001$). The mean measured $\delta^2H_v$ in water vapor was depleted by $2.89 \pm 2.26‰$ relative to the rain water $\delta^2H_{v,e}$ values. These biases were

magnified when calculating $d$ (Figure 3c). Derived equilibrium vapor $d_{v,e}$ values were relatively depleted by $7.8 \pm 3.08‰$.

It is well established that partial raindrop evaporation occurs below the cloud base because atmospheric humidity rarely achieves saturation through the entire depth over the course of an event (Lee et al., 2006). Partial raindrop evaporation acts to enrich the raindrop in heavy isotopes as the

lighter isotopes preferentially escape to the atmosphere due to kinetic fractionation (Stewart, 1975; Jacob and Sonntag, 1991). This is especially true for short duration and low magnitude convective rain events (Yu et al., 2006; Tian et al., 2007; Wen et al., 2010; Huang and Wen, 2014). Worden et al. (2007) concluded that 20 to 50% of rainfall evaporates near convective clouds over tropical locations, leading to strong isotopic signatures as observed from the Tropospheric Emission Spec-

trometer (TES).

The results shown here are similar to other field-based studies. Lee et al. (2006) concluded that observed $\delta^{18}O_v$ in water vapor and that derived from the equilibrium theory for a site in New Haven, Connecticut, USA agreed to within -2.5 to 1.5‰. Wen et al. (2010) reported that values for a site in Beijing China were within $-0.76 \pm 1.90‰$, $1.9 \pm 9.9‰$ and $7.7 \pm 8.3‰$ for $\delta^{18}O_v$, $\delta^2H_v$, and $d_x$

(uncertainty reported as 1 standard deviation), respectively. They demonstrated that the departures from the equilibrium values could be described as a linear function of relative humidity, with larger departures observed at lower humidities (i.e. suggesting greater raindrop evaporation). Our tall tower observations did not show a significant linear relation to relative humidity, although the differences between observations and the equilibrium calculation tended to decrease at relative humidity greater

than 95% and as the precipitation magnitude increased. Precipitation data collected from 2008 to 2011 near the tall tower site also support that isotope ratios in precipitation tend to be more enriched in heavy isotopes for small rainfall events. Overall, the difference between observed isotope ratios in water vapor and the equilibrium values are small and partial raindrop evaporation likely contributes to this observed bias.





### 3.2 Controls on isotope composition of water vapor

The relation between $\delta^{18}O_v$ and water vapor mixing ratio measured at 185 m (2010 to 2012) is compared with the three isotope models (RM1, RM2, and EM1 defined above) for different time periods (Figure 4) to gain further insights regarding the dominant processes influencing the tall tower observations. Given the large number of hourly water vapor observations ($n$ = 13,312), these data are displayed using a smoothed histogram technique (Eilers and Goeman, 2004). On an annual basis, an upper bound is defined by the simple two-source mixing model (EM1). A lower bound is defined by RM2 (a Rayleigh model that allows for a rain-out fraction of 30%). Assuming a simple closed system, RM1 provides an intermediate fit, and its curvature relative to the data density contours, illustrates that Rayleigh processes have a predominant influence on the oxygen isotope composition of the PBL vapor.

Given the initial conditions of the air mass, described above, the best fit Rayleigh model yielded an $r^2$ of 0.76 and an equilibrium fractionation factor of $\alpha = 1.0103$ ($p < 0.05$) (equivalent to a condensation temperature of 15 °C). Lee et al. (2006) also reported a large warm bias in the condensation temperature when applying the same type of model to their annual data set in New Haven, Connecticut, USA. The best fit Keeling mixing model yielded an $r^2$ of 0.37 and a very realistic estimate of the oxygen isotope composition of surface evaporation (-7.4‰, $p < 0.05$). Although the process of surface evaporation explained much less of the total variation in PBL vapor compared to the Rayleigh model, the relatively high coefficient of determination and statistical significance of the best fit parameters provides some evidence that surface evaporation within the region strongly modifies the oxygen isotope composition of vapor arriving at the tall tower.

Closer examination of the growing season data indicates that the rain out fraction may exceed $f = 30\%$ as evidenced by the relatively large isotope depletion that occurs for water vapor mixing ratios between 15 and 20 mmol mol$^{-1}$. It is also possible that these observations are associated with smaller convective summertime rain events when partial raindrop evaporation is favorable (Yu et al., 2006; Tian et al., 2007; Wen et al., 2010; Huang and Wen, 2014). The best fit Rayleigh and Keeling models explained 59 and 50% of the variation, respectively. During the non-growing season the best fit Rayleigh and Keeling models explained 72 and 28% of the variation, respectively. The density plot shows that curvature of the data is similar to the Rayleigh model, however, the highest data density region (see bright yellow shaded contours) indicate a departure from this curvature that is consistent with evaporation effects.

The tall tower vapor data differ substantially from the GMWL and the Local Meteoric Water Line (LMWL, $\delta^2H = 7.8\delta^{18}O + 6.9$) (Figure 4). The growing season PBL Water Vapor Line (WVL, $\delta^2H = 6.2\delta^{18}O - 9.01$, $r^2 = 0.90$, $p < 0.05$) yielded a relation that was inversely related to the local leaf water from agricultural plants ($\delta^2H = 2.7\delta^{18}O - 37.1$) and the soil ($\delta^2H = 5.3\delta^{18}O - 21.6$) indicating that leaves and soil were important sources of the PBL vapor. If the isotope composition of water vapor within the region were determined primarily by precipitation inputs (i.e. if the vapor





were in isotope equilibrium with precipitation) then the $\delta^2$H-$\delta^{18}$O relation would be equal to the LMWL. If we make this assumption, a growing season water vapor equilibrium line can be calculated (WVL$_{eq} = \delta^2$H $= 7.4\delta^{18}$O $- 0.18$). The slope and intercept of the WVL and WVL$_{eq}$ relations are

statistically different ($p < 0.05$ and $p < 0.1$) and demonstrate that under non-condensing conditions (i.e. $h < 100\%$), the isotope composition of water vapor is not simply derived from the precipitation, but is modified by other processes. Welp et al. (2008) came to a similar conclusion for field-scale measurements conducted within a few kilometers of the tall tower during the summer of 2006.

While the GMWL parameter values are determined primarily by the Rayleigh distillation effect,
deuterium excess values ($d_x = \delta^2$H $- 8\delta^{18}$O) in water vapor are largely governed by non-Rayleigh distillation processes (Gat and Airey, 2006). Here, we observed large positive $d_x$ in vapor for all months. The mean annual values were 33.5 ‰ with mean monthly values ($>40$‰) observed from November through February. The mean growing season $d_x$ value was 25.9‰. The mean monthly values showed negative relations with water vapor mixing ratio ($y = -0.98x + 43.6$, $r^2 = 0.55$), air
temperature ($y = -0.83x + 41.0$, $r^2 = 0.52$), and precipitation amount ($y = -0.09x + 39.3$, $r^2 = 0.37$), and a very weak positive relation with relative humidity ($y = 1.28x - 68.5$, $r^2 = 0.08$).

Based on an analysis of water vapor $d_x$ from several mid-latitude locations, Welp et al. (2012) found that the diurnal variability was likely controlled by two dominant processes including plant transpiration and PBL water vapor entrainment. Lai and Ehleringer (2011) also observed a strong
influence of PBL entrainment on the early morning variations in $d_x$ in a Pacific West Coast Douglas fir forest. Huang and Wen (2014) have also examined the factors controlling $d_x$ over cropland in Zhangye, northwest, China. In their analyses, they showed that variation in the deuterium excess of ET explained 94% of the variation in daytime water vapor $d_x$, implying that at some locations water vapor $d_x$ is an excellent tracer of ET. The recent work of Zhou et al. (2014) suggests that plant
transpiration has a dominant influence on vapor $d_x$ on diurnal timescales. At the longer timescales (monthly) examined here we expect that the variability and departure from the GMWL is influenced by synoptic conditions and air mass trajectories with strong modification by surface ET from within the region. For instance, the large $d_x$ values observed during the non-growing season, especially during November and December, suggest the important role of near surface water evaporation (i.e.
large kinetic fractionation effects) (Gat, 1996) within the region and probably reflect the dominant contributions of evaporation from bare agricultural soils and the Great Lakes, of which the latter reach peak evaporation rates in late fall and early winter (Blanken et al., 2011). During the main growing season, $d_x$ was less positive because plant transpiration is a non-discriminating process under equilibrium conditions (Zhou et al., 2014) and represents a substantial fraction of ET.

To further explore the influence of Rayleigh, ET, and PBL growth processes on the isotope composition of the PBL, we performed cross wavelet multivariate analyses for near continuous time series observed in August 2010 (Figure 5 and Figure 6). Analyses for the Rayleigh modeled oxygen isotope composition of water vapor ($\delta^{18}$O$_R$) versus the tall tower $\delta^{18}$O$_v$ observations (Figure 5)



demonstrate relatively strong in-phase coherence through the month of August 2010 across a broad

range of periods. It is interesting to note when the Rayleigh relation fails to describe the observations. For example, at periods greater than 64 hours and periods less than 8 hours there are numerous days in August 2010 when the Rayleigh relation and observations show little or no coherence. Identifying the exact mechanisms that account for these discrepancies is challenging because many meteorological processes operating in the PBL are not independent (i.e. there is feedback between ET and PBL

growth (McNaughton and Spriggs, 1986)). For example, Figure 6 shows there is strong coherence with a phase lag of about 3 hours (90 degrees) between ET and PBL growth rate for diurnal cycles (periods ranging from 8 to 32 hours) for nearly the entire month of August, 2010. Figure 5 also shows the wavelet coherence between the ET isoforcing and the time derivative of $\delta^{18}O_v$ as well as the PBL growth rate versus the time derivative of $\delta^{18}O_v$. These analyses show that there are a

number of more isolated periods when there is strong coherence, confirming that both ET and PBL growth are key forcing factors (Lee et al., 2012).

Similar analyses were also performed to examine the behavior of $d_x$ (Figure 6). These analyses reveal the influence of synoptic/air mass effects and PBL effects on $d_x$. For example, similar coherence was observed for wind direction versus $d_x$ and water vapor mixing ratio versus $d_x$. The coherence

was significant at synoptical scales (periods ranging from 100 to 256 hours or 4 to 10 days) implying the importance of synoptic scale air mass back trajectories. The effects of PBL growth and ET on $d_x$ clearly operate at different periods through the time series. The effects of PBL growth rate showed significant coherence at diurnal scales (periods ranging from 4 to 64 hours), while the ET showed significant coherence with $d_x$ on diurnal (8 to 32 hours) and synoptic (128 to 256 hours) scales. In

many cases, the phase lag between ET and $d_x$ implies that ET is leading the change in $d_x$.

To probe this further, we focus our attention on the ET isoforcing (oxygen isotope) characteristics (Figure 7). Using the WRF modeled PBL heights we estimated the ET isoforcing effect over the depth of the PBL for each hour. The time derivative of the ET isoforcing was then compared to the time derivative of $\delta^{18}O_v$. The time series and distributions of these derivatives show that they are of

similar magnitude. Here, the mean absolute values of both distributions indicate that ET can account for about 53% of the variation in $\delta^{18}O_v$ for August 2010 implying that ET is a dominant controlling factor.

A case study of high PBL water vapor concentration (defined here as $\geq 30$ mmol mol$^{-1}$) was carried out to further examine the underlying controlling factors. The extreme event of July 14,

2010 had a maximum dew point temperature of 26°C at 1300 LST. Local water vapor mixing ratios increased from about 22 to 39 mmol mol$^{-1}$ over the 24-hour period. The locally measured and modeled vapor fluxes were very high, ranging up to 10.6 mmol m$^{-2}$ s$^{-1}$ near midday. Over a 12-hour period, starting at midnight, we calculated the change in water vapor concentration within the PBL that was associated with the average rate of ET for the tall tower domain. These calculations indicate

that ET could account for about 8.4 mmol mol$^{-1}$ change (about 83% of the observed variation) in



the PBL water vapor concentration. The WRF-STILT source footprint analyses are shown for this case in Figure 8. These results illustrate that the vapor source was associated with NNE to ESE flow the day before (July 13, 2010) with flow switching to WNW the day after (July 15, 2010) the extreme event. The highest water vapor concentrations were observed on July 14, 2010 when the flow was southerly before the passage of a cold front. The source footprint intensity was greatest in Minnesota, Iowa, and Indiana and was dominated by agricultural sources (59%).

Additional evidence is provided by the tall tower isotope data and isoforcing (oxygen isotope) calculations. The tall tower observations during this period indicate that the $\delta^{18}O$ of water vapor increased steadily from about -18‰ to -13‰. Further, for the same 12-hour period as described above, the instantaneous $I_F$ averaged 0.08 m s$^{-1}$‰. Therefore, over the 12-hour period, the $I_F$ associated with ET accounted for a 3.8 ‰ variation in the PBL vapor and about 61% of the observed variation. Thus, multiple lines of evidence support that this extreme dew point event was substantially enhanced by local/regional evaporation. These observations also support the general relationship described below in Figure 9 indicating that a high fraction of the PBL water vapor was generated locally.

Although other approaches have been used to infer the impact of the US Corn Belt (Changnon et al., 2003) on regional humidity, the combined data, analytical, and modeling approaches used here offer a unique and more direct quantification. The higher amplitude of crop transpiration rates during the mid growing season indicate that summertime humidity can be significantly amplified by crops and may, therefore, enhance convective precipitation.

### 3.3 Evapotranspiration contribution to PBL vapor and Precipitation

WRF modeling and isotope mixing model analyses were used to help constrain the contribution of regional ET to PBL water vapor. The mean (2008-2011) growing season latent heat flux densities for each land use class within the study domain (i.e. the inner-most domain of 80 x 80 km) were approximately 25 (0.57), 114 (2.6), 119 (2.7), 112 (2.5), 130 (2.9), and 14 (0.32) W m$^{-2}$ (mmol m$^{-2}$ s$^{-1}$) for urban, dryland crops, dryland crops/grasslands, grasslands, evergreen needle leaf forest, and lakes, respectively. The area-weighted contribution of each land use type to the total evaporative flux for the study domain was dominated by dryland crop (58%) and dryland crops/grasslands (42%), respectively. The growing season contributions to evaporation for all other land use types were insignificant according to the WRF-NOAH modeling (and given the spatial resolution for the domain) over the period 2008 to 2011.

The WRF land use evaporation analysis was combined with the oxygen isotope observations using a simple mixing model to help constrain the relative contributions of ET to PBL water vapor. Since the area-weighted flux densities indicate that evaporation is dominated by the agricultural land use, we make use of the key isotope signals from the agricultural component and a simple two end-member isotope mixing model. Figure 9 shows the histograms of the fraction of local vapor ($f_v$),





estimated using the oxygen isotope mixing model for the daytime for June through August. These histograms indicate that median $f_v$ was 23%, 33%, and 37% during the 2010-2012 growing seasons, respectively. The fraction of local vapor is also plotted as a function of the PBL water vapor
mixing ratio observed at 185 m. The PBL vapor partitioning followed a saturation-type function $(f_v = 0.66\chi_w/(14.7 + \chi_w), r^2 = 0.18, p < 0.001)$. This relation indicates that the fraction of local water vapor increases asymptotically with water vapor mixing ratio. As expected, small changes in local evaporation can have a stronger effect on the fraction of water vapor in the PBL when mixing ratios are relatively low ($< 10$ mmol mol$^{-1}$). At mixing ratios of 25 mmol mol$^{-1}$, this relation
implies that the locally-generated vapor from evaporation accounts for about 42% of water vapor in the PBL. Also shown in Figure 9 is the fraction of PBL water vapor derived from ET as simulated by WRF for June to August, 2010. The WRF simulations indicate that on average daytime ET accounted for about $61 \pm 18\%$ of the PBL water vapor. The median water vapor mixing ratios in 2010, 2011, and 2012 were 19.7, 18.1 and 15.9 mmol mol$^{-1}$, respectively, indicating that the
locally generated vapor accounted for 38, 36, and 34% of the signal. Based on global analyses, best estimates indicate that approximately 40,000 km$^3$ of water vapor are transported to the continents each year, with ET from terrestrial ecosystems accounting for 73,000 km$^3$ (Trenberth et al., 2007b; Trenberth and Asrar, 2014). This global ratio of oceanic advection to terrestrial ET implies that 65% of the vapor signal over the continents is derived from ET and is considerably larger than our median
values obtained for the PBL in the Upper Midwest, United States.

     The different estimates of $\delta_{ET}$ provide a way of evaluating the relative uncertainty of the mixing model approach. For example, a change in the mean flux-weighted isotope composition of evaporation by +3‰ would shift the relations observed in Figure 9 lower. At mixing ratios of 25 mmol mol$^{-1}$ the local contributions to PBL water vapor would be lower by approximately 6%. Further, if
the isotope composition of the background vapor were 3‰ lower, the sensitivity of the partitioning approach to the background estimate of the isotope composition of vapor would shift the relation observed in Figure 9 higher. At mixing ratios of 25 mmol mol$^{-1}$ the local contributions to PBL water vapor would be higher by approximately 2%. This sensitivity is lower compared to changes in $\delta_{ET}$ because $\delta_b$ appears in the numerator and denominator of equation 10.

As described above, the isotope composition of the annual (non-growing and growing season) precipitation for the period 2006-2011 closely followed the GMWL. Here we examine in more detail the isotope composition of precipitation during the growing season to gain new insights regarding source origin and regional recycling. As discussed by Trenberth and Asrar (2014), numerical models tend to overestimate local-scale moisture recycling so that additional constraints provided by
empirical data may be used to help diagnose such biases.

     Examination of growing-season (May 1 to August 31) precipitation in $\delta^2$H-$\delta^{18}$O space indicated a near identical slope (8.04) to the GMWL, and a smaller intercept (8.3) with $r^2 = 0.94$. Figure 10 shows that $f_p$ ranged from close to 0 to 0.96 over the period, with a median value of 0.26. Inter-


estingly, Figure 10 indicates that from DOY 121 to DOY 180 that $f_p$ was approximately 0.10 and

increased significantly to 0.54 for the period DOY 180 to DOY 240. This step change is coincident
with high land surface evaporation during this period of peak growth for the agricultural region.
Further, it has been shown that the Great Plains Low Level Jet (GPLLJ) has a strong influence on va-
por transport into the region and can have an important effect on regional water recycling (Harding,
2014). Based on the model data presented by Harding (2014) (his Table 2.5, the 100 strongest warm

season precipitation events in the North Central U.S.) the median recycling ratio was 12.1% with a
range of 4.2 to 34.6%. We re-examined these data and found that the recycling ratio increased as the
GPLLJ weakened ($y = -0.099x + 0.18$, $r^2 = 0.18$) indicating that local ET becomes increasingly
important as long-distant transport from the Gulf of Mexico weakens.

Because the $\delta_{advx}$ and $\delta_{ETx}$ are highly variable and subject to considerable noise, we performed

a Monte-Carlo simulation to provide a more robust growing season estimate of $f_p$ based on the ob-
served precipitation data from 2006 to 2011 at the tall tower. Here we use the Monte-Carlo approach
to select values of $\delta_{advx}$ and $\delta_{ETx}$ based on the tall tower observations from 2010 to 2011. The
Monte-Carlo method selected median values within the 95% confidence intervals. One thousand
simulations were performed to evaluate equation 8 for each precipitation event from 2006-2011.

Figure 10 shows the frequency distribution of values. Notice that we did not filter any of the $f_p$
estimates so that there are a few values that fall outside of the realistic range. Overall, we find that
the growing season $f_p$ value was 0.31, indicating that terrestrial evaporation significantly enhances
the warm season precipitation.

Atmospheric water recycling is expected to be strongly linked to climate change with amplifica-

tion anticipated during wet periods (Vallet-Coulomb et al., 2008). Bosilovich and Schubert (2002)
used a general circulation model with water vapor tracers to follow their transport through the model
atmosphere. They concluded that 14% of the water precipitated within the US Midwest was derived
from local ET. Zangvil et al. (2004) restricted their numerical modeling analyses to the growing
season and U.S. Corn Belt and estimated that the water recycling index ranged up to 45%. In fact,

they found that seasonal and monthly analyses masked the importance of recycling on short (daily)
time scales. As discussed by Trenberth (1998) the calculation of water recycling using numerical
models is scale dependent. In his analysis, annual moisture recycling in the Mississippi basin was
on the order of 7% and up to 21% during the summertime when using a length-scale of 1800 km.
Further, Eltahir and Bras (1996) also suggest that summertime water recycling within the Missis-

sippi basin is on the order of 25%. Gat et al. (1994) used stable isotope analyses of precipitation to
estimate the contribution of evaporation from the Great Lakes to continental water vapor content.
In their study they estimated a contribution of 5 to 16%. These previous studies are in-line with our
own independent analyses and show that warm-season precipitation events have a relatively strong
local signature and that these rates are reasonably well-constrained by models at least on seasonal

time-scales.





## 4   Conclusions

1. The oxygen and hydrogen isotope composition of water vapor observed from a very tall tower in the Upper Midwest, United States shows a very strong seasonal amplitude ($\delta^{18}O_v$ = -40.1 to -15.5‰ and $\delta^2H_v$ = -278.7 to -109.1‰). The strong seasonal amplitude is driven by synoptic scale (Rayleigh) processes that are strongly modulated by planetary boundary layer processes including evapotranspiration and entrainment.

2. Isoforcing calculations support that evapotranspiration can have a dominant influence on the fluctuations of $\delta^{18}O_v$. Wavelet coherence analyses were used to demonstrate that the deuterium excess of water vapor is influenced by both synoptic and planetary boundary layer processes. Based on coherence and phase relationships it appears that changes in evapotranspiration often lead changes in deuterium excess.

3. Based on multiple lines of evidence (modeling and tall tower isotope observations), the humidification of the planetary boundary layer and the occurrence of extreme dew point temperatures have a strong terrestrial evaporation fingerprint. At water vapor mixing ratios greater than 25 mmol mol$^{-1}$ the locally-generated vapor from evapotranspiration accounts for 40 to 60% of the water vapor in the planetary boundary layer. Source footprint analyses for extreme dew point events indicate that the source of this evapotranspiration is largely ($\approx 90\%$) traceable to agricultural crops within the region.

4. The contribution of evapotranspiration to growing season precipitation (precipitation recycling ratio) was estimated using a simple isotope mixing model that was constrained using three years of tall tower isotope observations of water vapor and six years of isotope observations of precipitation. A Monte-Carlo analysis indicates that the precipitation recycling ratio is about 30% and in relatively good agreement with estimates derived from numerical weather models.

*Acknowledgements.* Funding for this research was provided by the Minnesota Corn Research and Promotion Council (4101-14SP). Support for the Rosemount, Minnesota AmeriFlux core site was provided by the U.S. Department of Energy's Office of Science. XL acknowledges support from the US National Science Foundation (grant 1520684). We thank Minnesota Public Radio and Tom Nelson for providing logistical support for the tall tower (KCMP) isotope observations. Finally, we acknowledge the support from the University of Minnesota Supercomputing Institute (MSI) for Advanced Computational Research.



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



**Table 1.** Water vapor and precipitation isotope climatology

| Month | $\chi_w$ [1] | $\delta^{18}O_v$ [2] | $\delta^2H_v$ [2] | $d_x$ [3] | Isof-$^{18}O$ [4] | Isof-$^2H$ [4] | $\delta_{ET}$ $^{18}O$ [5] | $\delta_{ET}$ $^2H$ [5] | $\delta^{18}O_P$ [6] | $\delta^{18}O_P$ [# 6] | $\delta^2H_P$ [6] | $d_{xP}$ [7] |
|---|---|---|---|---|---|---|---|---|---|---|---|---|
| Jan | 2.3 | -40.1 | -278.7 | 42.1 | 0.0019 | - | -22.5 | - | -22.4 | -21.0 | -162.4 | 5.6 |
| Feb | 3.0 | -34.6 | -232.4 | 44.4 | 0.0024 | - | -31.1 | - | -15.3 | -16.1 | -121.3 | 7.5 |
| Mar | 5.7 | -27.7 | -185.3 | 36.3 | 0.0002 | - | -25.2 | - | -9.9 | -10.3 | -67.7 | 14.7 |
| Apr | 6.3 | -25.0 | -170.8 | 29.2 | 0.0090 | - | -10.0 | - | -9.0 | -9.7 | -68.9 | 8.7 |
| May | 9.8 | -21.5 | -139.2 | 32.8 | 0.0066 | 0.0045 | -10.6 | -69.4 | -7.6 | -8.2 | -54.6 | 11.0 |
| Jun | 13.8 | -18.3 | -123.6 | 22.8 | 0.0086 | 0.0033 | -4.7 | -79.0 | -7.4 | -7.0 | -46.6 | 9.4 |
| Jul | 22.0 | -15.5 | -109.1 | 14.9 | 0.0053 | 0.0091 | -3.0 | -63.1 | -8.3 | -7.7 | -53.5 | 8.1 |
| Aug | 17.7 | -18.3 | -129.2 | 17.2 | 0.0041 | -0.0017 | -5.4 | -98.8 | -4.4 | -6.8 | -39.1 | 15.3 |
| Sept | 11.3 | -23.7 | -151.3 | 38.3 | 0.0071 | - | -6.2 | - | -8.5 | -8.8 | -57.6 | 12.8 |
| Oct | 7.6 | -25.1 | -162.4 | 38.4 | 0.0020 | - | -8.7 | - | -9.9 | -9.7 | -63.8 | 13.8 |
| Nov | 5.6 | -27.7 | -179.5 | 42.1 | 0.0029 | - | -19.0 | - | -8.0 | -12.3 | -80.3 | 18.1 |
| Dec | 4.9 | -35.9 | -243.3 | 43.9 | 0.0027 | - | -12.0 | - | -20.6 | -19.8 | -144.8 | 13.6 |
| | | | | | | | | | | | | |
| Mean | 9.2 | -26.1 | -175.4 | 33.5 | 0.0045 | 0.0038 | -13.2 | -77.6 | -10.9 | -11.5 | -80.1 | 11.6 |
| Min | 2.3 | -40.1 | -278.7 | 14.9 | 0.0019 | -0.0017 | -31.1 | -98.8 | -22.4 | -21.0 | -162.4 | 5.6 |
| Max | 22.0 | -15.5 | -109.1 | 44.4 | 0.0090 | 0.0091 | -3.0 | -63.1 | -4.4 | -6.8 | -39.1 | 18.1 |
| Range | 19.7 | 24.6 | 169.6 | 29.5 | 0.0071 | 0.0108 | 28.1 | 35.7 | 18.0 | 14.2 | 123.3 | 12.5 |

[1] Water vapor mixing ratios ($\chi_w$, mmol/mol) measured at 185 m and reported as median monthly values

[2] Water vapor isotope composition, $\delta^{18}O_v$ and $\delta^2H_v$ (permil) measured at 185 m and reported as median monthly values

[3] Deuterium excess of water vapor ($d_x$) was calculated from the median values $\delta^{18}O_v$ and $\delta^2H_v$ (permil)

[4] ET isoforcing calculations for the oxygen and deuterium isotope ratios (m s$^{-1}$ permil) are reported as median monthly values

[5] The oxygen and deuterium isotope flux ratio of ET ($\delta_{ET}$, permil) were derived from the tall tower gradient. Monthly values are flux-weighted by ET.

[6] Precipitation isotope composition $\delta^{18}O_P$ and $\delta^2H_P$ (permil) are reported as amounted weighted values. Precipitation related data were measured from 2006 to 2011 and monthly averages are also shown for the period 2010 to 2011 ($\delta^{18}O_P$)

[7] Deuterium excess of precipitation ($d_{xP}$, permil) was calculated from the monthly flux-weighted values

All vapor related data were measured at the tall tower from April 2010 to December 2012

Note that isoforcing and flux ratio values for deuterium are not reported for the non-growing season due to low signal to noise ratios.





**List of Figures**

**Figure 1.** Overview of research approach illustrating the tall tower location and study domain. A synthesis involving tall tower water vapor and isotope observations, field scale flux measurements, and numerical simulations were used to examine how evapotranspiration and planetary boundary layer processes influence water vapor and water recycling within the region.

**Figure 2.** Aircraft observations of the oxygen isotope composition of water vapor ($\delta^{18}O_v$) measured over a forested landscape in New England, USA (He and Smith, 1999, Table 2). Data from three campaigns show that $\delta^{18}O_v$ follows a powerlaw function (y = -32.1$\chi_w^{-0.213}$) of water vapor mixing ratio ($r^2$=0.98, $n$=24, $p$<0.0001).

**Figure 3.** Comparison of oxygen (left panel), hydrogen (middle panel), and deuterium excess (right panel) isotope composition of water vapor measured at 3 m and 185 m compared to the theoretical values for water vapor in isotope equilibrium with precipitation (falling rain drops) events during the 2010-2011 growing season. The solid lines show the 1:1 relation. The dashed lines show the best-fit regression.

**Figure 4**. Smoothed histogram plots of oxygen and deuterium isotope ratios in water. The left-hand panels illustrate oxygen isotope ratios in water vapor as a function of water vapor mixing ratio measured at a height of 185 m on the University of Minnesota tall tower. The right-hand panels show isotope ratios in water vapor, soil water, and local leaf water plotted in $\delta^{18}O$-$\delta^2H$ space. The lines illustrate different models and parametrizations (RM1, RM2, and EM1) as described in the text. The water vapor isotope data represent measurements taken from 2010 to 2012.

**Figure 5**. Wavelet coherence analysis of the oxygen isotope ratio of water vapor ($\delta^{18}O_v$) for August 2010. Hourly observations of water vapor mixing ratio and oxygen isotope ratio from the tall tower 185 m sample level (top left panel). Wavelet coherence of modeled oxygen isotope ratios using the best-fit Rayleigh model (described in the text) versus the observations (top right panel). Wavelet coherence of time derivative of $\delta^{18}O_v$ versus evapotranspiration isoforcing integrated over the depth of the PBL (bottom left panel). Wavelet coherence of time derivative of $\delta^{18}O_v$ versus PBL growth (bottom right panel). The color bar represents the local correlation coefficients in time-frequency space. The period is shown in hours. The black arrows represent the phase angle relationship between the variables. Arrows pointing east and west show signals that are in perfect phase and antiphase, respectively. Arrows pointing north show that variable 1 leads variable 2 (defined in figure titles) by a phase shift of 90 degrees.

**Figure 6**. Wavelet coherence analysis of deuterium excess ($d_x$) for August 2010. Hourly observations of water vapor mixing ratio and deuterium excess from the tall tower 185 m sample level (top left panel). Wavelet coherence of ET versus PBL growth (top right panel). Wavelet coherence of wind direction versus PBL growth (middle left panel). Wavelet coherence of water vapor mixing ratio versus deuterium excess (middle right panel). Wavelet coherence of PBL growth versus the time derivative of deuterium excess (bottom left panel). Wavelet coherence of ET versus the time derivative of deuterium excess (bottom right panel). The color bar represents the local correlation coefficients in time-frequency space. The period is shown in hours. The black arrows represent the phase angle relationship between the



variables. Arrows pointing east and west show signals that are in perfect phase and antiphase, respectively. Arrows pointing north show that variable 1 leads variable 2 (defined in figure titles) by a phase shift of 90 degrees.

**Figure 7**. The influence of ET isoforcing (oxygen isotopes) on the oxygen isotope composition of PBL water vapor during August 2010.  Hourly ET (mmol $m^{-2}$ $s^{-1}$) measured by the eddy covariance approach over agricultural crops located within the footprint of the University of Minnesota tall tower (top left panel). PBL height simulated using WRF3.5 for the tall tower location (top right panel). Tall tower ET isoforcing calculation (middle left panel). ET isoforcing calculation integrated with respect to PBL height and compared to the time derivative of the oxygen isotope ratio of water vapor ($\delta^{18}O_v$)  (middle right panel). Relative frequency distribution of the time derivative of $\delta^{18}O_v$ observations (bottom left panel). Relative frequency distribution of the integrated ET isoforcing calculations (bottom right panel).

**Figure 8**. Source footprint analysis of planetary boundary layer water vapor arriving at the University of Minnesota tall tower based on the Stochastic Time-Inverted Lagrangian Transport (STILT). These data and analyses represent a high dew point event that occurred on July 14, 2010.

**Figure 9**. Relative frequency distributions of PBL water vapor partitioning ($f_v$) for 2010 (top left panel), 2011 (top right panel), 2012 (middle left panel) and all data combined (middle right panel). The relative frequency distribution is also shown for estimates derived from the Weather Research and Forecasting (WRF3.5) model simulations for June-August, 2010 (lower left panel). Here, the average daytime values represent the fraction of water vapor in the PBL derived from local ET evaluated under the following conditions, ET > 0, and −udX/dx > 0 and −vdX/dy >0. The lower right panel shows the fraction of evaporated vapor contained in the planetary boundary layer as a function of total water vapor mixing ratio.

**Figure 10**. Precipitation recycling ratio estimated using a simple deuterium excess mixing model. The panels from top to bottom represent: Deuterium excess in precipitation; deuterium excess of water vapor measured at 185 m on the tall tower (i.e. approximation of the advection term); deuterium excess of evapotranspiration determined from the tall tower flux ratio method; Precipitation recycling ratio; and estimate of growing season precipitation recycling ratio for 2006-2011 based on precipitation and tall tower isotope data and a Monte Carlo simulation.



**Figure 1**

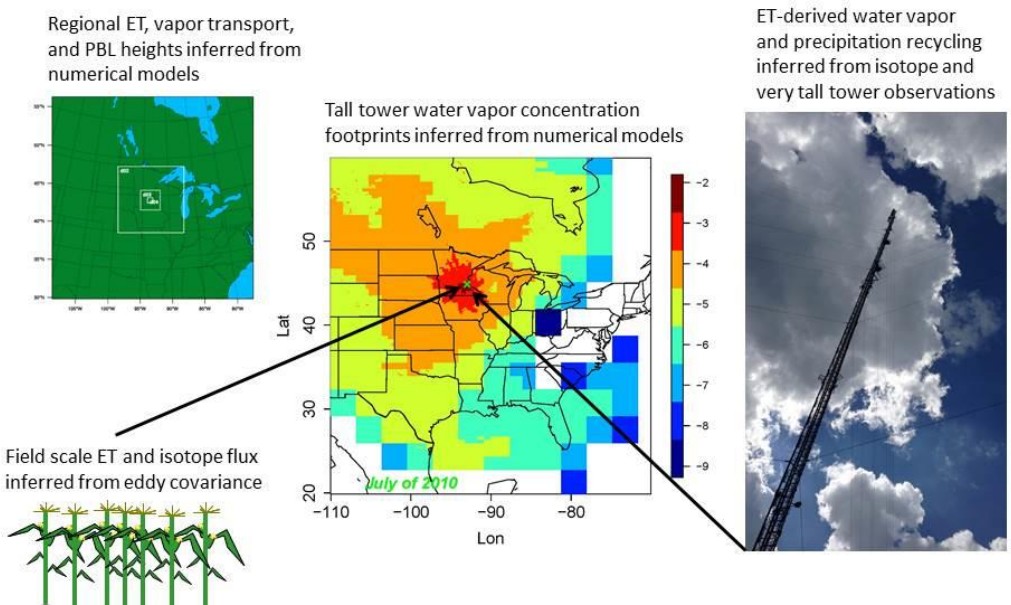




**Figure 2**

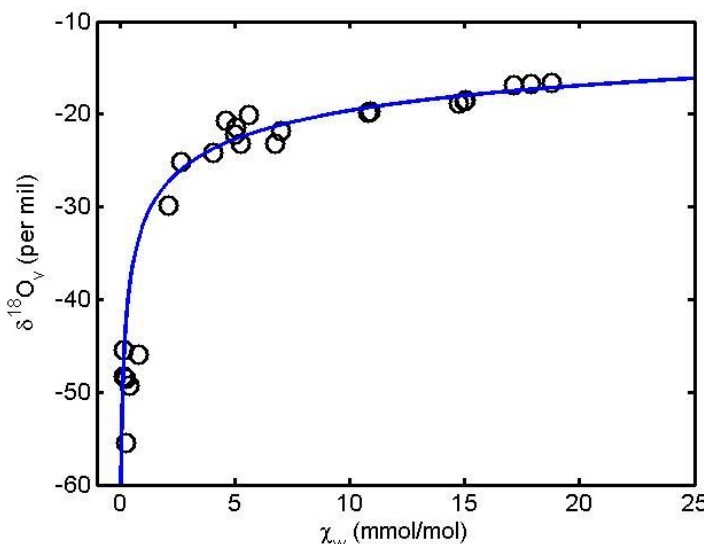





**Figure 3**

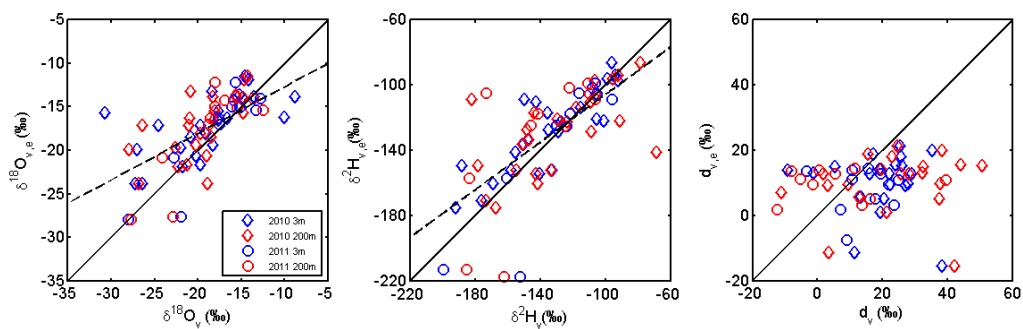





Figure 4

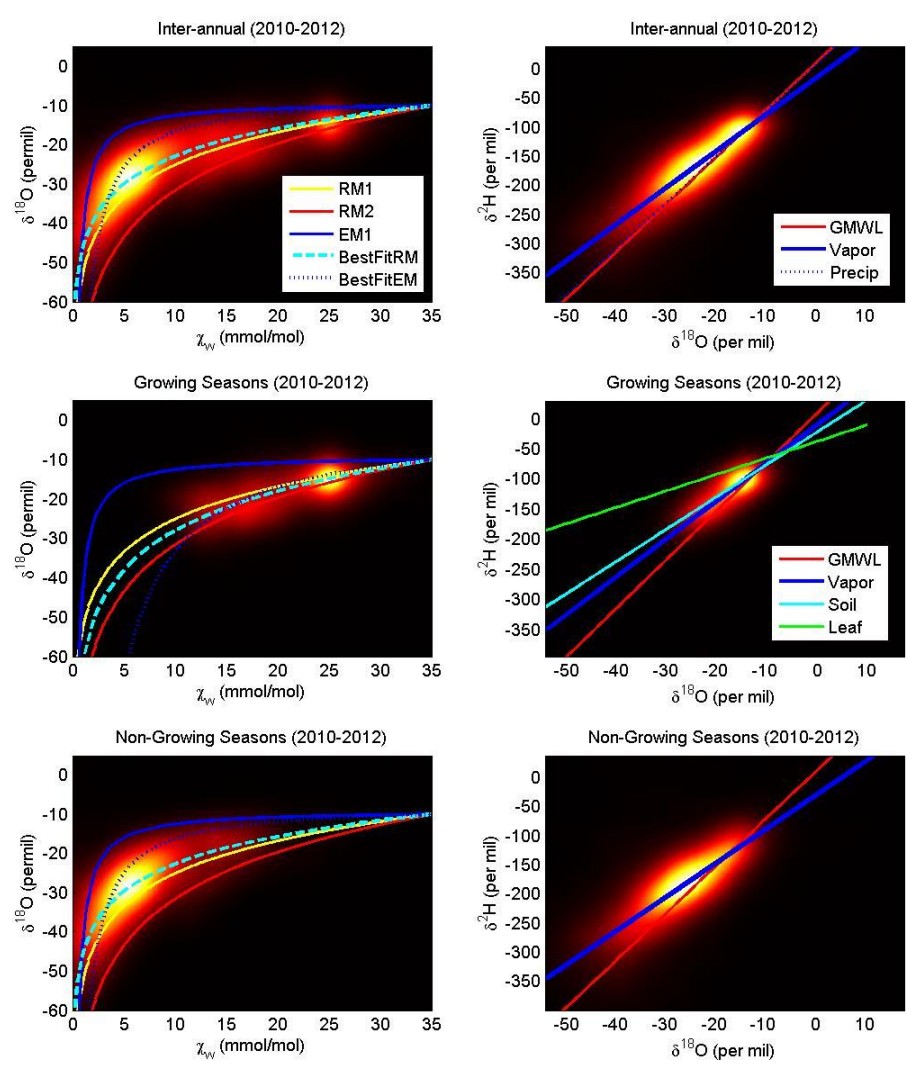





**Figure 5**

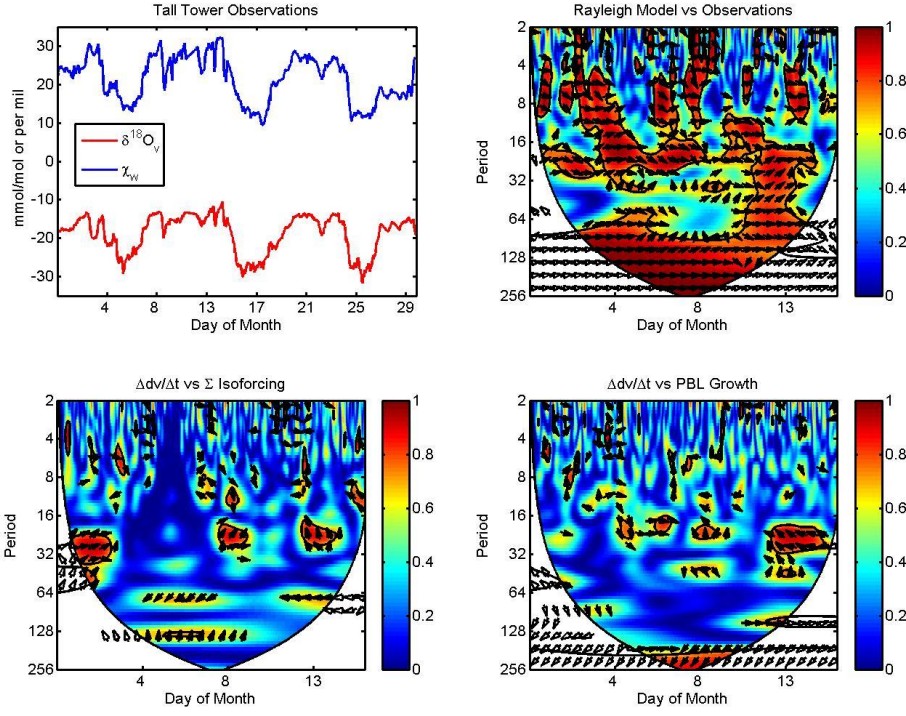





**Figure 6**

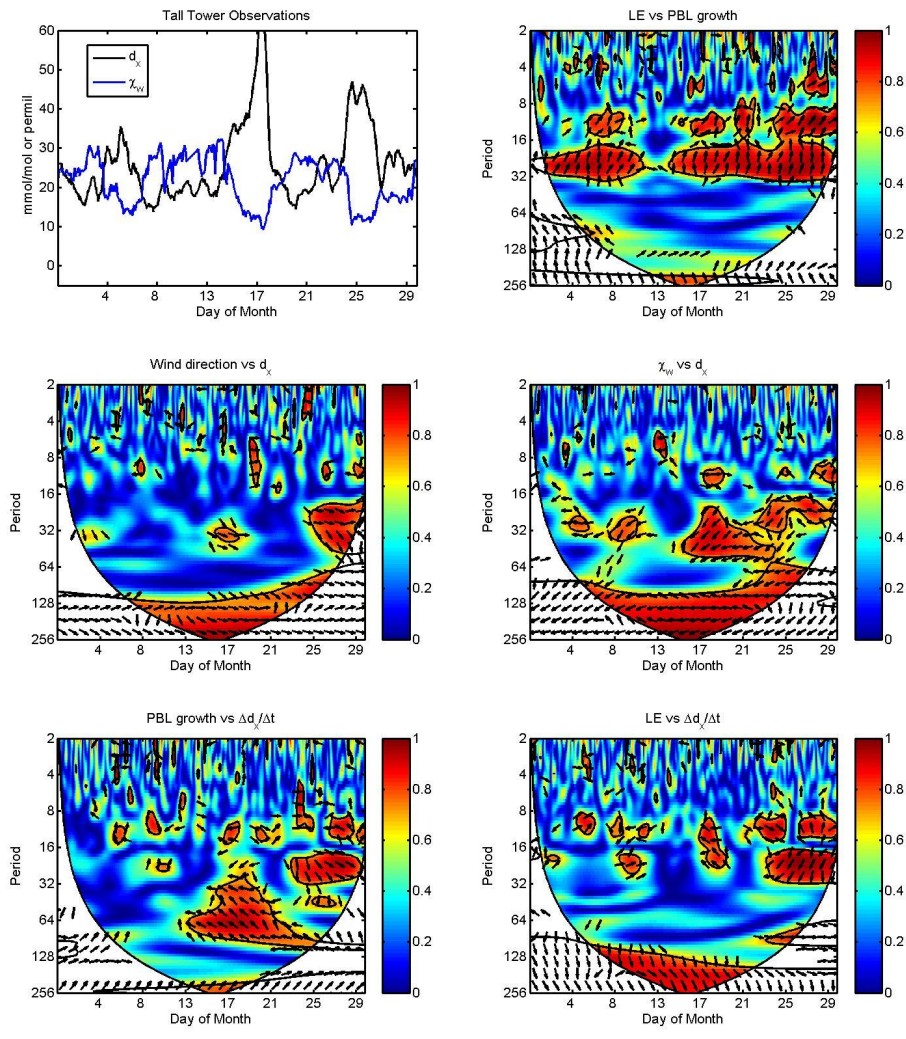



**Figure 7**

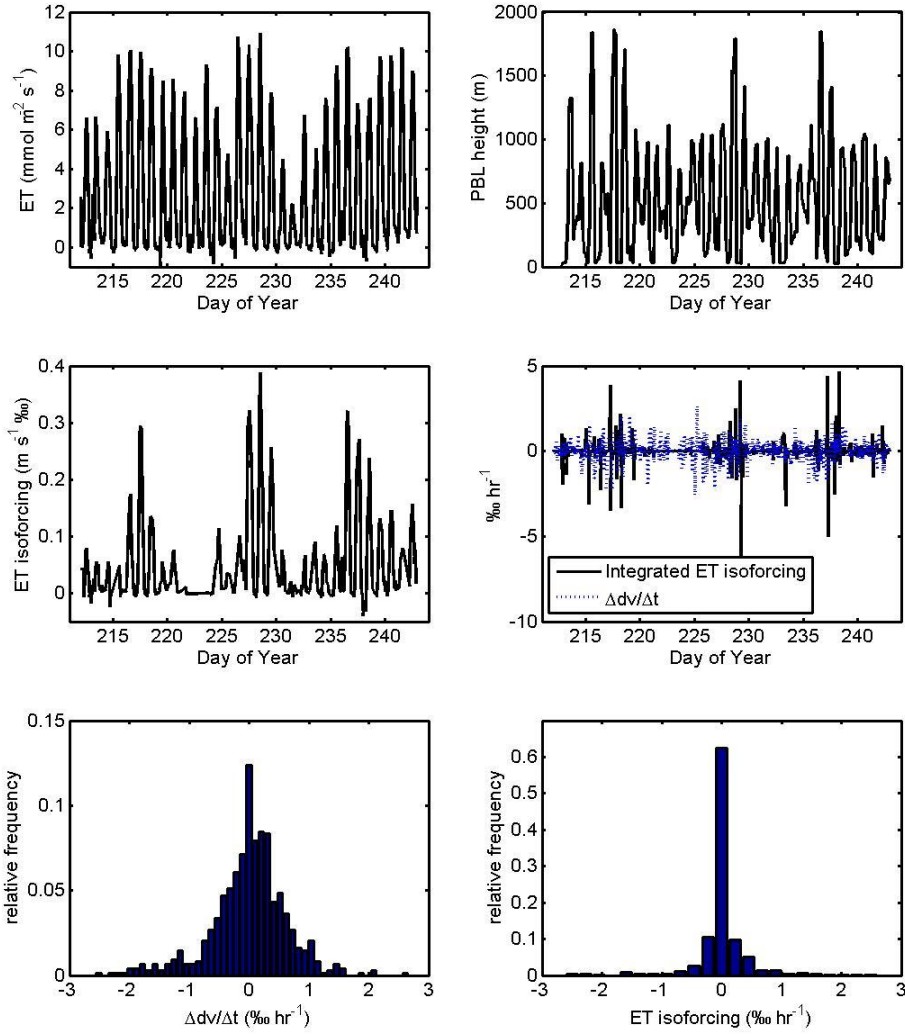




**Figure 8**

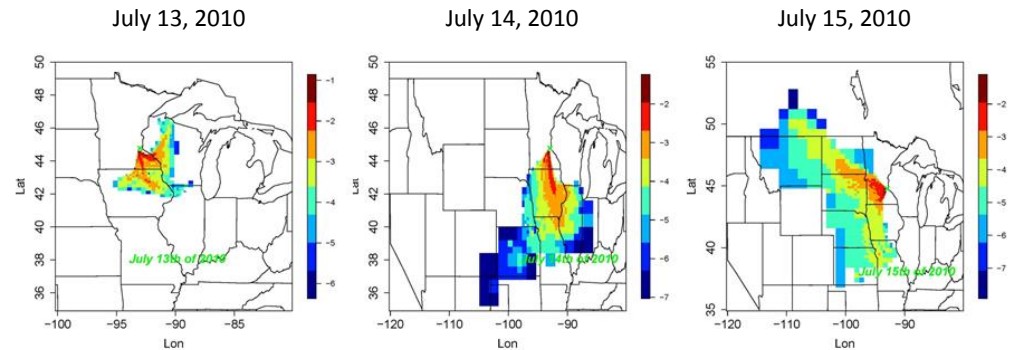




**Figure 9**

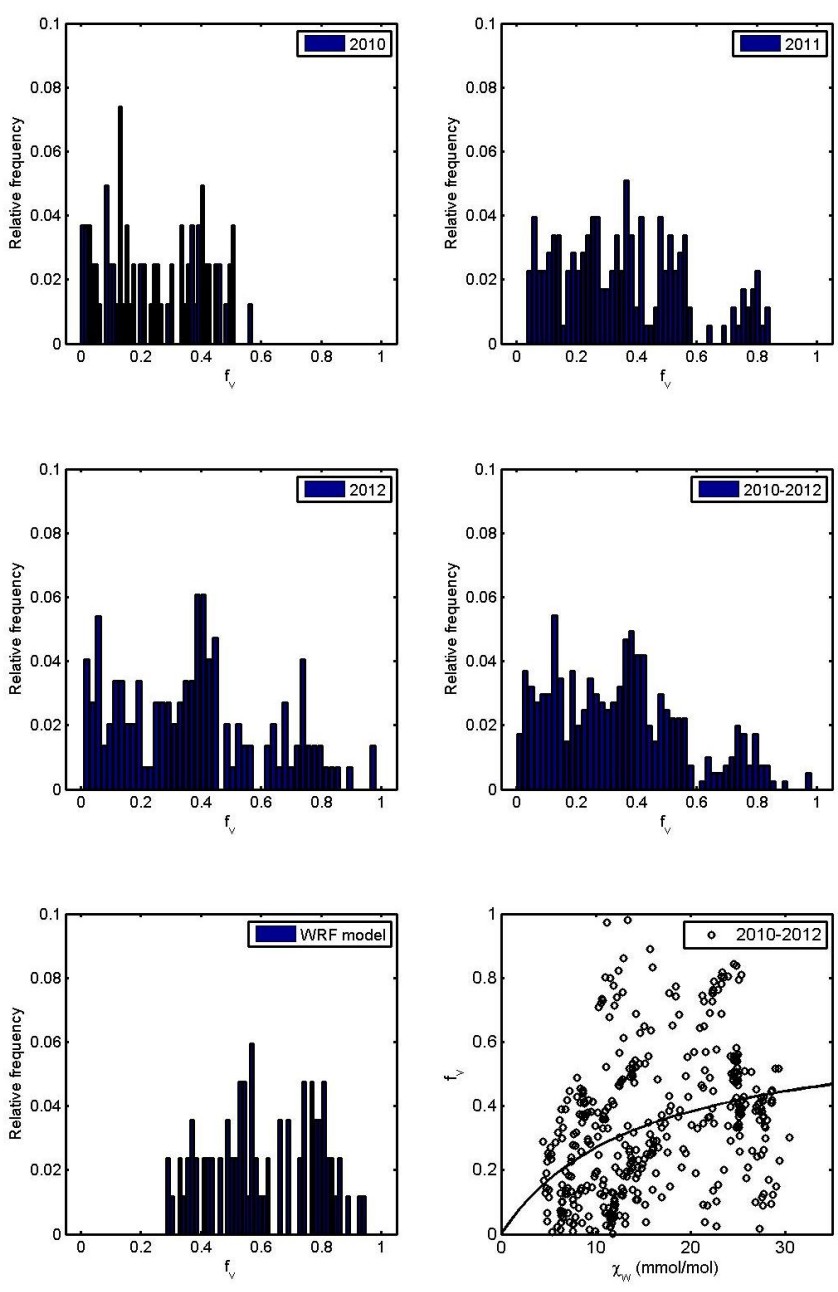



**Figure 10**

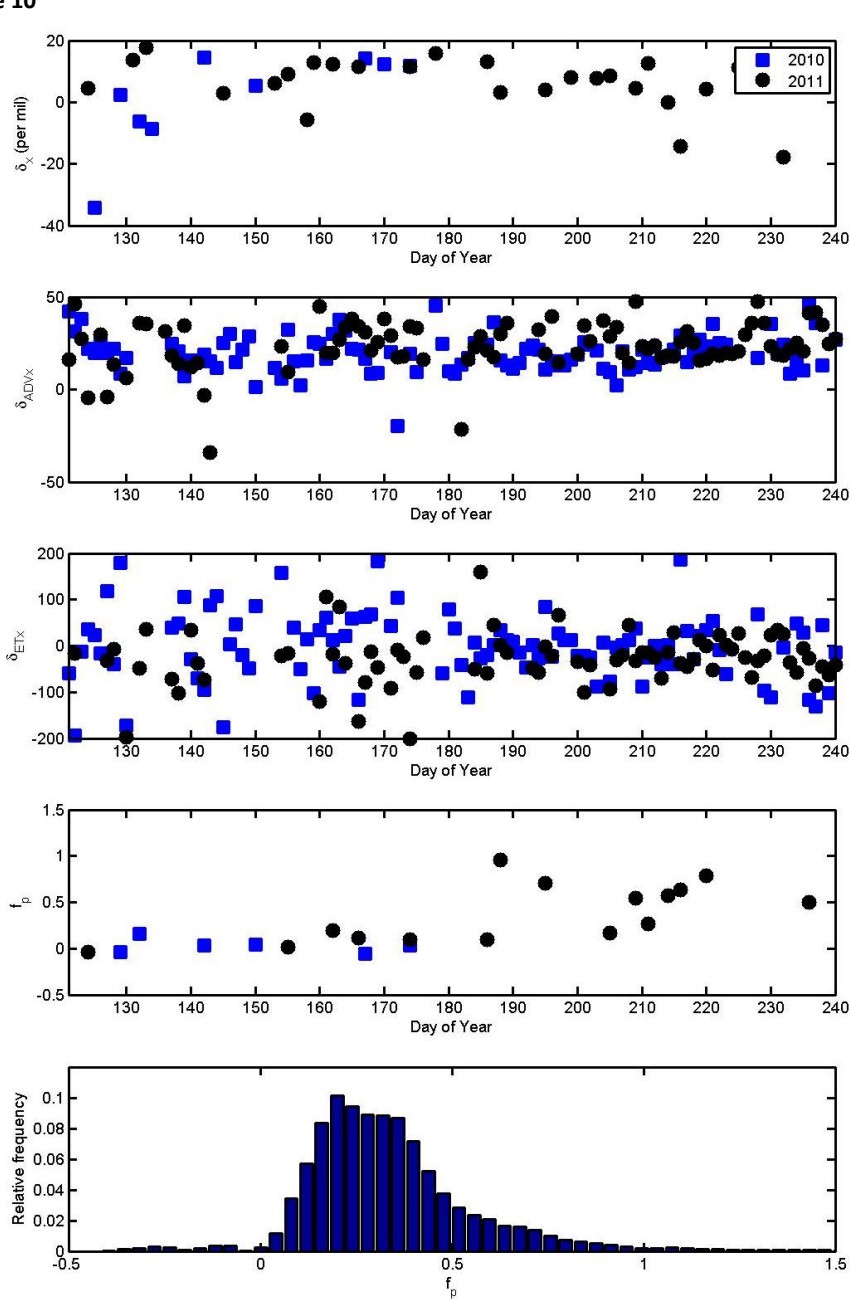