# Peer review of "Investigating the Source, Transport, and Isotope Composition of Water Vapor in the Planetary Boundary Layer"

_Atmospheric Chemistry and Physics, 2015_

## Referee Comment (RC1) · Anonymous Referee #1 · 29 Feb 2016

Review of

**"Investigating the Source, Transport, and Isotope Composition of Water Vapour in the Planetary Boundary Layer"**

by T. J. Griffis et al.

Paper published in ACPD on 18 January 2016

**1   General Comments**

This paper presents and discusses a comprehensive and very valuable dataset covering a three-year period (2010-2012) of hourly measurements in water vapour isotopes from a tall tower (inlets at $3\,\mathrm{m}$ and $185\,\mathrm{m}$) in the planetary boundary layer (PBL) as well as event-based precipitation samples in the Upper Midwest, United States. The isotope signature of evapotranspiration was also determined from the tall tower flux-gradient measurements. The authors analyse the data using several simple isotope models to describe the processes occurring in idealised airmasses (1) due to condensation of an initial airmass of oceanic source (called RM1), (2) due to rain out of the condensate (called RM2) and (3) due to mixing between surface evaporate and near-surface atmospheric water vapour (called EM1). Estimates of "regional" evapotranspiration and of the recycling ratio of precipitation during the growing season are provided using a simple two end member mixing model. One selected month of hourly data is analysed in more detail with respect to the coherence of behaviour of different variables (vapour isotopes from the tall tower predicted isotope composition of the PBL with a simple Rayleigh model, Evapotranspiration, boundary layer growth, temporal change of the isotope signals) using wavelet coherence spectra. Relevant questions with respect to the continental water cycle and the processes driving the isotope variability in the planetary boundary layer are discussed. I recommend publication of this manuscript after the following points have been addressed:

A **Presentation of the large amount of data:**

  1. The water vapour isotope and precipitation climatology presentation would be easier to follow if a Figure with 3 different panels ($\delta^{18}$O, $\delta^2$H, $d$) would show the vapour, precipitation and ET annual cycles.

  2. Many Figures and Table 1 should be made clearer, see my specific comments below.

  3. Additional Tables might be necessary to help the reader particularly, when data from older studies is used or assumptions are made for the use of the simple isotope models (see specific comments below).

B **Application of the simple isotope models and numerical modelling for determining boundary layer heights and evapotranspiration:**

  1. A short motivation for the modelling approach (use of the simple isotope models as well as for the numerical modelling to obtain PBL heights) should be given in the introduction. Why are these models used? (p. 3, L. 65)

  2. It should be made clearer during the presentation of the models in the methods how the "BestFitRM" and the "BestFitEM" are obtained and what they represent.

C **Regional recycling ratios:** When discussing the "origin" of boundary layer water vapour throughout the manuscript it is not clear what is meant quantitatively/geographically by "regional" evapotranspiration. The same is true for the presented precipitation recycling fractions. The authors mention on p. 17, L. 567 that "As discussed by Trenberth, 1998 the calculation of water recycling using numerical models is scale dependent". This is also true for empirical recycling ratios and moisture origin quantifications. A statement on what the term "regional" means should therefore be given in the paper.

**2  Specific comments**

1. p. 1, L. 9: Instead of "Models" I would suggest "Several simple isotope models".

2. p. 1, L. 9: I would avoid to use the term "Rayleigh process" in the abstract. Maybe "rainout along an idealised airmass trajectory" or at least "Rayleigh distillation process" would be clearer.

3. p. 1, L. 13: "at the annual time-scale" is confusing, only hourly and monthly mean data are discussed here, maybe "when looking at the full hourly dataset" would be clearer.

4. p. 2, L. 50-51: An other study presenting 5 months of deuterium excess measurements in the continental PBL is Aemisegger et al. 2014, ACP.

5. p. 3, L. 63: "Rayleigh" should be explained here.

6. p. 3, L. 74: I do not understand hypothesis 2, why would one assume this a priori?

7. p. 3, L. 61: $d_x$ should be changed into $d_v$ for vapour, $d_p$ for precipitation,... to be consistent with the $\delta^{18}O_v$ notation used.

8. p. 4, L. 98: What was the pumping rate and tubing type used?

9. p. 4, L. 103: The isotope composition of the span values should be mentioned.

10. p. 4, L. 103: The sampling time is relatively short, what was the response time of the system?

11. p. 4, L. 112: No parenthesis for the reference to Griffis et al., 2010b

12. p. 7, L. 206: The reference to Lee et al. 2005 seems inadequate here for the definition of the liquid water isotope ratio.

13. p. 7, L. 220: "Three simple models...": it should be mentioned here that the parameters for the models are chosen from "best guesses" to compare their predictions to the measurement data and that best fits of the parameters to the measurement data are subsequently calculated. It should also be mentioned how the fitting is done (i.e. which parameters are fitted) for the best fit lines in Figure 4. Furthermore, it would help the reader if a Table would summarise the parameters of the models and their assumed numerical values.

14. p. 8, L. 223: How was the condensation temperature of -3°C determined?

15. p. 8, L. 231: "where precipitation/condensation is evaporated" I do not understand this. Souldn't it say "where precipitation/condensate is removed"?

16. p. 8, L. 237: Why is $\delta^{18}O = -7.7‰$ used for $\delta_{ET}$? Is this just a best guess?

17. p. 8, L. 243: Is $\delta_v$ from 185 m height used here? How independent are $\delta_{ET}$ and $\delta_v$

18. p. 8, L. 250: This approach neglects changes in $\delta$ by advection. This should be mentioned.

19. p. 8, L. 253: It should be explicitly mentioned that substantial changes in the background $\delta_v$ depending on the synoptic scale weather situation can occur.

20. p. 9, L. 265-277: I do not understand the role of this paragraph. On L. 237 a value of $\delta^{18}O = -7.7‰$ is used. And there are the tall tower $\delta^{18}O_{ET}$ presented in Table 1. This is confusing for me.

21. p. 9, L. 277: The reference is missing.

22. p. 9, L. 280: Equation 11, shouldn't $d$ be used here instead of $\delta$?

23. p. 9, L. 283: What are the uncertainties of the $d_{ET}$ estimated from the tower measurements?

24. p. 11, L. 227: "biased low", shouldn't it be "high"? And I do not agree with the terminology "biased", which I find confusing. As the authors mention on L. 332 below-cloud evaporation can strongly affect the isotope composition of boundary layer vapour and precipitation. See also other experimental studies combining vapour and precipitation measurements: Risi et al. 2008, QJRMS, Tremoy et al. 2012, GRL, Aemisegger et al. 2015, GRL.

25. p. 11, L. 331: "depleted" can be confusing here. "$d_{v,e}$ were ... lower than $d_v$" would be better.

26. p. 12, L. 367: "warm bias" with respect to what?

27. p. 12, L. 383: add "the" before "curvature"

28. p. 12, L. 385: which panel in Figure 4 are the authors referring to? This last sentence of the paragraph (L.281-285) is not very clear to me.

29. p. 13, L. 419: "near surface water evaporation" is "soil evaporation" meant here?

30. p. 13, L. 420-425: very similar observations were made using PBL vapour deuterium excess in Europe (Aemisegger et al. 2014, ACP)

31. p. 13, L. 427: It is not clear which Rayleigh model was used here.

32. p. 14, L. 458: The hourly time series of the relevant variables for this case study should be shown in a Figure.

33. p. 14, L. 464: What is meant by "the tall tower domain"?

34. p. 15, L. 483: Does this mean that if bare soil instead of crop was there we would expect weaker convective precipitation purely from a moisture availability point of view? I am not sure I understand what "enhance" convective precipitation means here. Compared to which alternative scenario?

35. p. 15, L. 490: It would help to have a table for the latent heat fluxes of the different land use classes.

36. p. 16, L. 505: The PBL vapour partitioning should be discussed more critically. The correlation between the fraction of local water vapour and the local water vapour mixing ratio is relatively low and the scatterplot in Figure 9 looks very noisy.

37. Table 1 would be much clearer if the units were indicated in the individual columns. It is not clear what the two $\delta^{18}O_p$ exactly refer to. They seem to cover different time periods but it is not clear which is which. The additional information added below the table should be placed in the legend. Standard deviations of the hourly data for each month would be informative of the isotope signals' variability in the different seasons. Why is the water vapour isotope data at $3\,m$ not shown?

38. Figures in general: it would help the reader a lot if more and clearer references to the Figures were given in the text and alphabetical numbering of panels was used.

39. Figure 4: the colorbar is missing, it is not clear how BestFitRM, EM are obtained, it would help the reader if the chosen colormap would be white in the low density domain. Some of the lines are not easy to distinguish.

40. The role of Figure 10 is not clear to me.

---

## Referee Comment (RC2) · C.-T. Lai (Referee) · 2 Mar 2016

This manuscript presents a 3-year (2010-2012) dataset of stable isotope ratios in PBL water vapor ($\delta$) sampled from multiple inlets on a tall tower in a crop field in the Upper Midwest United States. Precipitation isotope composition (2006-2011) also was presented. The main objectives of the study are 1) to investigate water sources and fluxes that contribute to the variability in mixing ratios and isotope ratios of water vapor in PBL, 2) using a land surface model and wavelet analyses to discern the importance of PBL processes (Rayleigh distillation, evapotranspiration (ET) and tropospheric entrainment) on $\delta$, 3) to quantify ET contribution to PBL water vapor content and 4) to estimate recycling ratio of precipitation. Major findings of this manuscript include: 1) a dominant

influence of ET on $\delta$ and deuterium excess (dx), 2) a considerable contribution by local ET to PBL moisture content in the growing season (>40%) and 3) a 30% precipitation recycling ratio during growing season for this region.

The debate over uncertainties in the isotopic mass balance calculation for the global estimate of plant transpiration to continental water fluxes (T/ET) seems to come to an agreement (Jasechko et al. 2013, Good et al. 2015, Evaristo et al. 2015). However, uncertainties in the importance of ET to region/local precipitation and atmospheric moisture content remain high and field studies that aim to reduce this uncertainty is welcome and should be encouraged. This paper tackles this knowledge gap with numerical modeling and simple two source mixing calculation of isotopic information. Conclusions are drawn from multiple-lines of results. This work will be a valuable contribution to the isotope hydrology community. I recommend publication with minor revisions. My comments aim to improve the presentation of this work, and are detailed in the following:

Ln 117&118, provide model # and manufacturers for tipping bucket rain gauge and snow board. Ln179 what is rationale in choosing NOAH (a simple bucket) land surface model? Is your calculation sensitive to choice of land surface scheme?

Ln223 is the condensation temperature of -3 oC at the lifted condensation level site specific? Is it set as a constant for the entire study period?

Ln231 it reads awkward to say 'condensation is evaporated'; replacing precipitation/condensations by raindrops/condensates?

Ln236 be consistent with your definition of $\delta$ET; evaporation and evapotranspiration are used interchangeably throughout the text

Ln242 here $\delta$ET is determined using tall-tower flux-gradient approach, rather than a value of -7.7‰ from eddy covariance measurements. Why? How different are the two estimates?

[Figure]

Ln245 please explain why atmospheric gradients are considerably small for 2H

Ln263 This is an interesting approach. What is the vapor mixing ratio at 700hPa for the entire study period? This is important, as the power law function is hypersensitive at low mixing ratios.

Ln278 The authors state that "Following the methodology of Gat et al. (1994)", a two member mixing model approach (Eq 11) was used to estimate the recycling ratio of precipitation. However, the method described in Gat et al. (1994) was quite different than a two member mixing model. The authors need to clarify the claim, and explain the basis of Eq (11). Why is deuterium excess used in this calculation, instead of delta ratios?

Ln300 What does this agreement mean? What is its significance?

Ln302 Do you mean uncertainty or range in the precipitation isotope data?

Ln303 Shouldn't $\delta$ET (-77.6‰ be relatively "enriched" compare to precipitation (-80‰ ln289)?

Ln310 It is very interesting that the authors chose to compare measurements made in a crop field in upper Midwest of USA to measurements made in an urban setting in China. If local ET play a major role in regional PBL water vapor (as claimed here), why would you expect the two studies comparable? I do not see the relevance of this comparison.

Ln319-354 The whole discussion on the discrepancy between observed $\delta$v and that calculated from equilibrium assumption with precipitation weakens the manuscript in my opinion. Equilibrium only holds true when h=100%. As the authors correctly pointed out, reasons (mechanisms) behind this discrepancy has been well established elsewhere (e.g., the citations given). Measurements of isotopic composition of water vapor are rare before spectroscopy analyzers became commercially available. Hence, isotopic composition of water vapor was commonly assumed in equilibrium with precipitation (readily available) in the old days. The manuscript is better off without this comparison. I suggest the discussion here be shortened.

Ln345-350 Cut out these sentences; equilibrium should only be considered when h approaches 100%!

Ln388 delete "yielded a relation that". How was WVL and leaf/soil water "inversely related"?

Ln390 just say ET rather than "leaves and soil"

Ln395-396 Duh! equilibrium should only be considered when h approaches 100%! Consider deleting ln 390-398 for reasons stated above.

Ln406, the very weak positive relation is in contrast with the negative d-rh relationship found over seawater and at some continental sites. Discuss the possible causes for the discrepancy.

Ln510, Report errors for the 42% estimate. Calculated fv has a very large spread around the best-fit function.

---

## Author Comment (AC1) · 4 Apr 2016

**Response to review of Atmospheric Chemistry and Physics, MS No.: acp-2015-923**

Title: Investigating the Source, Transport, and Isotope Composition of Water Vapor in the Planetary Boundary Layer
Author(s): T. J. Griffis et al.
MS No.: acp-2015-923
MS Type: Research article
* * *
We thank the Reviewers for their thoughtful comments, criticisms, and attention to detail. All of the reviewer comments have been addressed below (review query in Italic; our response in blue upright Times New Roman).

**Reviewer #1**

*1 General Comments*

*This paper presents and discusses a comprehensive and very valuable dataset covering a three-year period (2010-2012) of hourly measurements in water vapour isotopes from a tall tower (inlets at 3m and 185m) in the planetary boundary layer (PBL) as well as event-based precipitation samples in the Upper Midwest, United States. The isotope signature of evapotranspiration was also determined from the tall tower flux- gradient measurements. The authors analyse the data using several simple isotope models to describe the processes occurring in idealised airmasses (1) due to condensation of an initial airmass of oceanic source (called RM1), (2) due to rain out of the condensate (called RM2) and (3) due to mixing between surface evaporate and near-surface atmospheric water vapour (called EM1). Estimates of "regional" evapotranspiration and of the recycling ratio of precipitation during the growing season are provided using a simple two end member mixing model. One selected month of hourly data is analysed in more detail with respect to the coherence of behaviour of different variables (vapour isotopes from the tall tower predicted isotope composition of the PBL with a simple Rayleigh model, Evapotranspiration, boundary layer growth, temporal change of the isotope signals) using wavelet coherence spectra. Relevant questions with respect to the continental water cycle and the processes driving the isotope variability in the planetary boundary layer are discussed. I recommend publication of this manuscript after the following points have been addressed:*

We thank the Reviewer for their positive comments.

*A) Presentation of the large amount of data:*

1. *The water vapour isotope and precipitation climatology presentation would be easier to follow if a Figure with 3 different panels ($\delta^{18}O$, $\delta^{2}H$, d) would show the vapour, precipitation and ET annual cycles.*
2. *Many Figures and Table 1 should be made clearer, see my specific comments below.*
3. *Additional Tables might be necessary to help the reader particularly, when data from older studies is used or assumptions are made for the use of the simple isotope models (see specific comments below).*

As requested, we have added new figures to describe the annual cycles (time series) of the key data including water vapor, precipitation, and their isotope ratios. We have also included a time series of evaporation and isoforcing data for reference. However, since we already have a lengthy list of figures (10), we have added these figures to the supplemental materials since they present the raw data that are summarized in Table 1.

Regarding Part A points 2 and 3 above, we have addressed these comments below in the detailed reviewer comments section.

*B) Application of the simple isotope models* and numerical modelling for determining boundary layer heights and evapotranspiration:

*1. A short motivation for the modelling approach (use of the simple isotope models as well as for the numerical modelling to obtain PBL heights) should be given in the introduction. Why are these models used? (p. 3, L. 65)*

As recommended, we have provided a brief justification for the simple isotope and regional weather model used in this study. However, we believe this material is better suited for the methodology, where we introduce the various isotope models and numerical modeling.

This material has been added to page 6 beginning after line 190

"The NOAH land surface scheme option was selected for all WRF simulations for three reasons: First, it has been used extensively in the literature; 2) we have been using WRF-NOAH to forecast ET for our region and have tested it extensively against tall tower eddy covariance flux observations; and 3) the WRF-NOAH system is computationally efficient compared to other options such as WRF-CLM (Community Land Model surface scheme option)."

and page 8 line 135

"Three simple models were used to aid the interpretation of the tall tower $\delta^{18}O_v$ data. These models were selected because their physics are well understood and they represent three idealized processes that are thought to dominate the behavior of water vapor in the PBL (Worden et al., 2007; Lee et al., 2006)."

*2. It should be made clearer during the presentation of the models in the methods how the "BestFitRM" and the "BestFitEM" are obtained and what they represent.*

The description of these models and the fitting procedures used has been described in greater detail. Please see page 9 line 260.

"Finally, we optimized the RM1 and EM1 models to determine the equilibrium fractionation factor and the isotope composition of ET, respectively. These optimized models are referred to as BestFitRM and BestFitEM, respectively. These models were fit to the observed tall tower data using a non-linear fitting algorithm (fitnlm) implemented using Matlab (Matlab Version 2013b, The Mathworks Inc., Natick, Massachusetts, USA)."

*C) Regional recycling ratios: When discussing the "origin" of boundary layer water vapour through-out the manuscript it is not clear what is meant quantitatively/geographically by "regional" evapotranspiration. The same is true for the presented precipitation recycling fractions. The authors mention on p. 17, L. 567 that "As discussed by Trenberth, 1998 the calculation of water recycling using numerical models is scale dependent". This is also true for empirical recycling ratios and moisture origin quantifications. A statement on what the term "regional" means should therefore be given in the paper.*

The concentration footprint of an active scalar such as water vapor is difficult to quantify. Based on our past research we have shown that the tall tower carbon dioxide and nitrous oxide concentrations are representative of the US Corn Belt. However, because water vapor has a relatively short life-time we take a conservative approach an estimate its concentration footprint on the order of 100 km. Therefore, our tall

tower observations and the inner-most domain for the WRF-NOAH numerical modeling (80 km x 80 km) are used here to define the "regional signal". This region has similar land use, topography, and air temperatures. We therefore assume that isotope ratios of precipitation measured at our site representative.

Therefore, to be clear, we have defined region as ~ 80 km x 80 km in reference to Figure 1, which introduces the study domain in the introduction. We also elaborate on this definition in the methodology, where we describe the concentration footprint of the tall tower.

*2 Specific comments*

*1. p. 1, L. 9: Instead of "Models" I would suggest "Several simple isotope models".*

We have edited this line as suggested to… "Several simple isotope models…."

*2. p. 1, L. 9: I would avoid to use the term "Rayleigh process" in the abstract. Maybe "rainout along an idealised airmass trajectory" or at least "Rayleigh distillation process" would be clearer.*

For clarity we have edited this line as suggested to… "the Rayleigh distillation process"

*3. p. 1, L. 13: "at the annual time-scale" is confusing, only hourly and monthly mean data are discussed here, maybe "when looking at the full hourly dataset" would be clearer.*

For clarity we have edited this line as suggested to … "when considering the entire hourly dataset"

*4. p. 2, L. 50-51: Another study presenting 5 months of deuterium excess measurements in the continental PBL is Aemisegger et al. 2014, ACP.*

Thank you for reminding us of this paper. We have revisited this work and have referenced it.

*5. p. 3, L. 63: "Rayleigh" should be explained here.*

As suggested we have explained what we mean by "Rayleigh" and have added the following text…

"Here, we examine the temporal scales and extent to which Rayleigh distillation (i.e. the removal of water vapor from the air mass *via* condensation and precipitation), evaporation, and PBL growth processes influence the isotope compositions ($\delta^2H_v$, $\delta^{18}O_v$, and $d_v$) of mid-continental atmospheric water vapor as observed in the Upper Midwest United States."

*6. p. 3, L. 74: I do not understand hypothesis 2, why would one assume this a priori?*

This hypothesis evolved from our survey of the literature where a number of studies have shown evidence that evapotranspiration can have a strong influence on $d_v$ especially in the late afternoon (after the PBL height has reached ~ steady-state). See publications by Lai et al., 2010, Oecologia, 165, 213-223 and Huang and Wen, 2014, JGR-Atmospheres, 119, 11,456-11,476.

*7. p. 3, L. 61: dx should be changed into dv for vapour, dp for precipitation,... to be consistent with the $\delta^{18}O_v$ notation used.*

As suggested, we have made this change here and throughout the manuscript including the Figures and Tables.

*8. p. 4, L. 98: What was the pumping rate and tubing type used?*

A large diaphragm pump (1023-101Q-SG608X, GAST Manufacturing Inc., Benton Harbor, Michigan, USA) pulled air continuously down the 185 m and 3 m sample inlets at a flow rate of approximately 3 L/min. The tall tower sample tubing is Synflex (Synflex Type 1300, Aurora, OH, USA). This type of tubing has been applied in other water vapor isotope applications (Wen et al., 2008; Lee et al., 2009; Wen et al., 2010; Welp et al., 2013).

We have edited page 4 line 100 to include these details.

*9. p. 4, L. 103: The isotope composition of the span values should be mentioned.*

Over the duration of the three-year measurement period the isotope composition of the dripper source water and therefore span values, was maintained at approximately -8.5 and -61.0 permil for $\delta^{18}O$ and $\delta^2H$, respectively. We have added this information to the methodology section on page 4 line 115.

As described in the methods, our calibration approach dynamically tracks the ambient water vapor mixing ratio by adjusting the amount of span water vapor mixed with ultra-dry air, while the isotope ratio is held constant. This calibration approach, as applied in the literature, has been reviewed by Griffis, 2013.

*10. p. 4, L. 103: The sampling time is relatively short, what was the response time of the system?*

Given the low pressure of the sub-sample inlets (40 kPa) and tunable diode laser sample cell (0.8 kPa) the equilibration time of the system was relatively fast, on the order of 5 seconds for the span calibrations and 30 seconds for the zero calibration. The TDL system response time has been described in detail in Griffis et al., (2010).

These details have been added to methodology on page 4 line 115-120.

*11. p. 4, L. 112: No parenthesis for the reference to Griffis et al., 2010b*

This has been corrected as suggested.

*12. p. 7, L. 206: The reference to Lee et al. 2005 seems inadequate here for the definition of the liquid water isotope ratio.*

To be more complete we have added fundamental references to Majoube (1971) and Jouzel, (1986).

Majoube, M. (1971), Fractionnement en oxygene-18 et en deuterium entre l'eau et sa vapeur, Journal de Chimie Physique, 10, 1423.

Jouzel, J., 1986: Isotopes in cloud physics: Multiphase and multistage condensation processes. *Handbook of Environmental Isotope Geochemistry,* B. P. Fritz and J. C. Foutes, Eds., Vol.2, Elsevier, 61–112

*13. p. 7, L. 220: "Three simple models...": it should be mentioned here that the parameters for the models are chosen from "best guesses" to compare their predictions to the measurement data and that best fits of the parameters to the measurement data are subsequently calculated. It should also be mentioned how the fitting is done (i.e. which parameters are fitted) for the best fit lines in Figure 4. Furthermore, it would help the reader if a Table would summarize the parameters of the models and their assumed numerical values.*

As recommended, we have provided more details regarding the model parameters and the parameter estimation approach on page 9 line 260. Because we report the best fit model parameters and statistics in the results and discussion for each panel in Figure 4, we have decided not to present them independently in a Table.

*14. p. 8, L. 223: How was the condensation temperature of -3°C determined?*

As a first approximation we used the mean annual surface air temperature for the site and adjusted it adiabatically given a typical PBL height of approximately 1.2 km.

*15. p. 8, L. 231: "where precipitation/condensation is evaporated" I do not understand this. Shouldn't it say "where precipitation/condensate is removed"?*

We have corrected this error and changed it to… "where precipitation/condensate is removed".

*16. p. 8, L. 237: Why is $\delta^{18}O = -7.7‰$ used for $\delta_{ET}$? Is this just a best guess?*

Originally we used this value from earlier eddy covariance isotope work as a first best guess. However, we now use a value of -6.2‰, which is the growing season value observed at the tall tower. We have modified page 10 line 305 accordingly.

*17. p. 8, L. 243: Is $\delta v$ from 185m height used here? How independent are $\delta ET$ and $\delta v$?*

Yes that is correct. The calculation of $\delta_{ET}$ involves the difference between the 185 m and 3 m levels and is therefore dependent on $\delta v$. However, the main point of the isoforcing calculation is to isolate the influence of ET on the PBL $\delta v$. This theory and experimental approach have been well established in Lee et al., (2009) – Global Biogeochemical Cycles, 23, GB1002.

*18. p. 8, L. 250: This approach neglects changes in $\delta$ by advection. This should be mentioned.*

As suggested we have noted that this approach neglects changes in $\delta$ by advection. Please see page 9 line 280,

*19. p. 8, L. 253: It should be explicitly mentioned that substantial changes in the background $\delta v$ depending on the synoptic scale weather situation can occur.*

As suggested we have noted that the "background" $\delta v$ can change substantially depending on synoptic scale weather patterns. Please see page 9 line 280.

*20. p. 9, L. 265-277: I do not understand the role of this paragraph. On L. 237 a value of $\delta^{18}O = -7.7‰$ is used. And there are the tall tower $\delta^{18}O ET$ presented in Table 1. This is confusing for me.*

Thank you for pointing this out. The main point of page 9 lines 265-277 is to show that we have multiple constraints on the isotope composition of evapotranspiration, which is a difficult variable to quantify. However, we have now added to the discussion our tall tower estimates of $\delta^{18}O_{ET}$ and have applied those values (mean growing season value of -6.2 ‰) in the modeling analysis. Please see page 10 line 300

*21. p. 9, L. 277: The reference is missing.*
Thank you. This has been corrected.

*22. p. 9, L. 280: Equation 11, shouldn't d be used here instead of δ?*

Thank you for catching this mistake. All of the terms in equation 11 should be *d* to represent deuterium excess.

*23. p. 9, L. 283: What are the uncertainties of the dET estimated from the tower measurements?*

We estimated the daytime $d_{ET}$ to range by up to 475 permil for the entire growing season. The uncertainty in $d_{ET}$ is relatively large primarily because the flux-gradient of $\delta^2H$ is very noisy. The standard error about the mean growing season $d_{ET}$ was 5.7 permil.

*24. p. 11, L. 327: "biased low", shouldn't it be "high"? And I do not agree with the terminology "biased", which I find confusing. As the authors mention on L. 332 below-cloud evaporation can strongly affect the isotope composition of boundary layer vapour and precipitation. See also other experimental studies combining vapour and precipitation measurements: Risi et al. 2008, QJRMS, Tremoy et al. 2012, GRL, Aemisegger et al. 2015, GRL.*

Thank you for pointing out this problem and catching the discrepancy related to the regression analysis. To avoid confusion, we have eliminated the term "bias" and simply refer to the key differences shown in Figure 3. Further, we have added some additional discussion related to these results by referring to the other papers suggested by the Reviewer.

Please see the revised text on page 12.

*25. p. 11, L. 331: "depleted" can be confusing here. "dv,e were ... lower than dv" would be better.*

This has been corrected as suggested.

*26. p. 12, L. 367: "warm bias" with respect to what?*

Here we are referring to the fact that the condensation temperature of an air mass for this location would be well below the value ($15^{o}C$) that was estimated from the optimization routine.

*27. p. 12, L. 383: add "the" before "curvature"*

This has been corrected as suggested.

*28. p. 12, L. 385: which panel in Figure 4 are the authors referring to? This last sentence of the paragraph (L.381-385) is not very clear to me.*

We have now explicitly referred to Figure 4a (the top left panel), which has also been labeled as suggested in previous comments.

*29. p. 13, L. 419: "near surface water evaporation" is "soil evaporation" meant here?*

We elaborate on page 13 line 420-421 that we are referring to soil evaporation and evaporation from the Great Lakes, of which the latter reach peak evaporation rates in the late fall and early winter.

*30. p. 13, L. 420-425: very similar observations were made using PBL vapour deuterium excess in Europe(Aemisegger et al. 2014, ACP)*
We have made detailed reference to the Aemisegger et al., 2015 paper here. Please see Page 14 line 455.

*31. p. 13, L. 427: It is not clear which Rayleigh model was used here.*

We used Rayleigh model RM2 as shown in Figure 4. This has been noted on page 15 line 465.

*32. p. 14, L. 458: The hourly time series of the relevant variables for this case study should be shown in a Figure.*

We currently have a relatively long list of figures (10). Given that there are 3 new figures presented, including Figure S1 that illustrates much of the raw data, we believe that this requested figure would have substantial repetition and would have limited new information content.

*33. p. 14, L. 464: What is meant by "the tall tower domain"?*

Here we are referring to the inner most model domain (80 km x 80 km) as described in the numerical modeling section. We have explicitly referred to the inner most domain on page 16 line 500.

*34. p. 15, L. 483: Does this mean that if bare soil instead of crop was there we would expect weaker convective precipitation purely from a moisture availability point of view? I am not sure I understand what "enhance" convective precipitation means here. Compared to which alternative scenario?*

During the mid-growing season the evapotranspiration from crops far exceeds the native vegetation (prairie savannah) that was replaced by agriculture. Because we have acquired recent data related to this, we have added a new figure that illustrates the high rates of evapotranspiration from crops relative to prairie. These sites are less than 2 km apart.

*35. p. 15, L. 490: It would help to have a table for the latent heat fluxes of the different land use classes.*

In an early draft of the manuscript we had presented this information as a figure and then removed it prior to submission. We have added this Figure to highlight how latent heat flux varies for the different land use classes in the supplemental information.

*36. p. 16, L. 505: The PBL vapour partitioning should be discussed more critically. The correlation between the fraction of local water vapour and the local water vapour mixing ratio is relatively low and the scatterplot in Figure 9 looks very noisy.*

We have provided a more critical review of Figure 9 and discussed the uncertainties. Please see Page 17 line 560. We have also added uncertainty bounds to Figure 9.

*37. Table 1 would be much clearer if the units were indicated in the individual columns. It is not clear what the two $\delta^{18}Op$ exactly refer to. They seem to cover different time periods but it is not clear which is which. The additional information added below the table should be placed in the legend. Standard deviations of the hourly data for each month would be informative of the isotope signals' variability in the different seasons. Why is the water vapour isotope data at 3m not shown?*

As requested we have revised Table 1 to make it clearer and have added other statistics to increase the information content. For clarity we only show one time period for the precipitation data (2010 to 2011) as these data provide the best seasonal coverage. We did not show the 3 m data because it follows very closely the seasonal pattern observed at 185 m. Further, we have now shown all of the precipitation isotope data for 2006 to 2011 in the supplemental information.

*38. Figures in general: it would help the reader a lot if more and clearer references to the Figures were*

*given in the text and alphabetical numbering of panels was used.*

We have edited the Figures so that each panel is labeled using alphabetical numbering (i.e. Figure 3a, 3b, etc)

*39. Figure 4: the colorbar is missing, it is not clear how BestFitRM, EM are obtained, it would help the reader if the chosen colormap would be white in the low density domain. Some of the lines are not easy to distinguish.*

We have edited the Figure to improve the clarity and added a colorbar.

*40. The role of Figure 10 is not clear to me.*

Because of the inherent uncertainties associated with the partitioning, we thought a Monte Carlo simulation would be a useful technique for understanding the overall uncertainty and robustness of our partitioning approach.

---

## Author Comment (AC2) · 4 Apr 2016

**Response to review of Atmospheric Chemistry and Physics, MS No.: acp-2015-923**

Title: Investigating the Source, Transport, and Isotope Composition of Water Vapor in the Planetary Boundary Layer
Author(s): T. J. Griffis et al.
MS No.: acp-2015-923
MS Type: Research article
* * *
We thank the Reviewers for their thoughtful comments, criticisms, and attention to detail. All of the reviewer comments have been addressed below (review query in Italic; our response in blue upright Times New Roman).

**Reviewer #2**

*This manuscript presents a 3-year (2010-2012) dataset of stable isotope ratios in PBL water vapor (δ) sampled from multiple inlets on a tall tower in a crop field in the Upper Midwest United States. Precipitation isotope composition (2006-2011) also was presented. The main objectives of the study are 1) to investigate water sources and fluxes that contribute to the variability in mixing ratios and isotope ratios of water vapor in PBL, 2) using a land surface model and wavelet analyses to discern the importance of PBL processes (Rayleigh distillation, evapotranspiration (ET) and tropospheric entrainment) on δ, 3) to quantify ET contribution to PBL water vapor content and 4) to estimate recycling ratio of precipitation. Major findings of this manuscript include: 1) a dominant influence of ET on δ and deuterium excess (dx), 2) a considerable contribution by local ET to PBL moisture content in the growing season (>40%) and 3) a 30% precipitation recycling ratio during growing season for this region. The debate over uncertainties in the isotopic mass balance calculation for the global estimate of plant transpiration to continental water fluxes (T/ET) seems to come to an agreement (Jasechko et al. 2013, Good et al. 2015, Evaristo et al. 2015). However, uncertainties in the importance of ET to region/local precipitation and atmospheric moisture content remain high and field studies that aim to reduce this uncertainty is welcome and should be encouraged. This paper tackles this knowledge gap with numerical modeling and simple two source mixing calculation of isotopic information. Conclusions are drawn from multiple-lines of results. This work will be a valuable contribution to the isotope hydrology community. I recommend publication with minor revisions.*

We thank the Reviewer for their positive comments.

*My comments aim to improve the presentation of this work, and are detailed in the following:*

*Ln 117&118, provide model # and manufacturers for tipping bucket rain gauge and snow board.*

We have added this information as requested. The tipping bucket rain gauge used was model 6028-B, All Weather Inc., CA, USA, and the snow board was provided by the Minnesota State Climate Office (http://climate.umn.edu/doc/journal/snowboard.doc).

*Ln179 what is rationale in choosing NOAH (a simple bucket) land surface model? Is your calculation sensitive to choice of land surface scheme?*

Our group has been working with the WRF-NOAH system since 2012 and has been testing it as an evaporation/irrigation forecasting tool (http://www.biometeorology.umn.edu/research/etool). While more detailed surface scheme options could be used such as WRF-CLM (Community Land Model surface scheme option), they are computationally much more expensive. From our experience the WRF-NOAH

system performs very well for our study domain. The Figure below (not shown in the manuscript) illustrates a comparison of modeled evaporation versus our tall tower eddy flux (185 m) estimates during the growing season. Based on these comparisons we are comfortable with the NOAH application in this current study.

[Figure]

*Ln223 is the condensation temperature of -3 °C at the lifted condensation level site specific? Is it set as a constant for the entire study period?*

Yes. The goal here is to simply provide an idealized case for comparison to the data. Please see response above to Review #1 who had a similar question.

*Ln231 it reads awkward to say 'condensation is evaporated'; replacing precipitation/condensations by raindrops/condensates?*

Thank you. Reviewer 1 also caught this mistake and it has been corrected.

*Ln236 be consistent with your definition of δET; evaporation and evapotranspiration are used interchangeably throughout the text*
Thank you for catching this. As recommended by Brutsaert (1982) in his book "Evaporation into the Atmosphere", we believe it is best to use the term evaporation and have, therefore, eliminated our usage of evapotranspiration. This gives consistent terminology throughout the manuscript.

*Ln242 here δET is determined using tall-tower flux-gradient approach, rather than a value of -7.7‰ from eddy covariance measurements. Why? How different are the two estimates?*

To be consistent, we now use the tall tower flux gradient values for all calculations. The growing season value was -6.2 ‰ and in close agreement with other previous approaches at our site. Because this variable is difficult to quantify, and is inherently noisy, the other estimates from eddy covariance, field-scale flux gradient, and analysis of xylem water are described for context and to provide an independent constraint on this key variable.

*Ln245 please explain why atmospheric gradients are considerably small for 2H*

This is simply due to the fact that the source strength is much smaller for deuterium. Therefore, for similar atmospheric mixing conditions, the gradient of deuterium in water vapor will be smaller compared to the gradient in the heavy oxygen isotope of water vapor. We have added a brief discussion to this effect on page 9 line 270.

*Ln263 This is an interesting approach. What is the vapor mixing ratio at 700hPa for the entire study period? This is important, as the power law function is hypersensitive at low mixing ratios.*

Over the three-year period the water vapor mixing ratio at 700 hPa was 3.9 mmol/mol. During the growing season (three-year period) the water vapor mixing ratio at 700 hPa was 5.9 mmol/mol. With respect to the power law function, these mean values occur before the function reaches its vertical asymptote (i.e. where it becomes hypersensitive). However, as shown in Figure 2, there are cases where the uncertainty in the background value will be large because of this sensitivity.

We have added the essentials of this discussion to Page 10 line 295.

*Ln278 The authors state that "Following the methodology of Gat et al. (1994)", a two member mixing model approach (Eq 11) was used to estimate the recycling ratio of precipitation. However, the method described in Gat et al. (1994) was quite different than a two member mixing model. The authors need to clarify the claim, and explain the basis of Eq (11). Why is deuterium excess used in this calculation, instead of delta ratios?*

Here we made an incorrect reference. We followed the methodology of Kong et al., 2013, published in Tellus B, 65, Art. 19251. We have adopted their equation 7.

*Ln300 What does this agreement mean? What is its significance?*

At relatively long times scales (seasonally) we would expect there to be isotope mass balance between inputs (precipitation) and outputs (evaporation, runoff, drainage). These data and analyses indicate that our atmospheric measurements provide a reasonable constraint on the isotope composition of evaporation, which is challenging to measure. We have added some addition discussion related to this point on page 11 line 335.

*Ln302 Do you mean uncertainty or range in the precipitation isotope data?*

Here we are referring to the overlap of uncertainty in the flux and precipitation values. We have clarified this in the text.

*Ln303 Shouldn't δET (-77.6‰ be relatively "enriched" compare to precipitation (-80‰ ln289)?*

Yes. Thank you for bringing this to our attention. There was a typo in Table 1.

*Ln310 It is very interesting that the authors chose to compare measurements made in a crop field in upper Midwest of USA to measurements made in an urban setting in China. If local ET play a major role in regional PBL water vapor (as claimed here), why would you expect the two studies comparable? I do not see the relevance of this comparison.*

We agree with the reviewer and have deleted this text.

*Ln319-354 The whole discussion on the discrepancy between observed $\delta v$ and that calculated from equilibrium assumption with precipitation weakens the manuscript in my opinion. Equilibrium only holds true when h=100%. As the authors correctly pointed out, reasons (mechanisms) behind this discrepancy has been well established elsewhere (e.g., the citations given). Measurements of isotopic composition of water vapor are rare before spectroscopy analyzers became commercially available. Hence, isotopic composition of water vapor was commonly assumed in equilibrium with precipitation (readily available) in the old days. The manuscript is better off without this comparison. I suggest the discussion here be shortened.*

As suggested, we have shortened this section and deleted the sentences noted below by Reviewer 2. Because this comparison represents one of the few ways to assess data quality, we wish to retain the essential elements presented.

*Ln345-350 Cut out these sentences; equilibrium should only be considered when h approaches 100%!*

Deleted as suggested.

*Ln388 delete "yielded a relation that". How was WVL and leaf/soil water "inversely related"?*

We have re-worked this section and deleted "yielded a relation that" and deleted "inversely related".

*Ln390 just say ET rather than "leaves and soil"*

We changed this to "evaporation" as suggested.

*Ln395-396 Duh! equilibrium should only be considered when h approaches 100%! Consider deleting ln 390-398 for reasons stated above.*

We have modified this section and simply show that these relations are statistically different.

*Ln406, the very weak positive relation is in contrast with the negative d-rh relationship found over seawater and at some continental sites. Discuss the possible causes for the discrepancy.*

Because the relationship is weak and not statistically significant we are reluctant to elaborate on a potential physical explanation for it at this time.

*Ln510, Report errors for the 42% estimate. Calculated fv has a very large spread around the best-fit function.*

We now show the uncertainty bounds in Figure 9 and report the 1 sigma uncertainty range as 21 to 62%.